# CONCEPT BOTTLENECK MODELS UNDER LABEL NOISE

## ABSTRACT

Concept bottleneck models (CBMs) are a class of interpretable neural network models that make the final predictions based on intermediate representations known as concepts. With these concepts being human-interpretable, CBMs enable one to better understand the decisions made by neural networks. Despite this advantage, we find that CBMs face a critical limitation: they require additional labeling efforts for concept annotation, which can easily increase the risk of mislabeling, *i.e.*, CBMs tend to be trained with noisy labels. In this work, we systematically investigate the impact of label noise on CBMs, demonstrating that it can significantly compromise both model performance and interpretability. Specifically, we measure the impact of varying levels of label noise across different training schemes, through diverse lenses including extensive numerical evaluations, feature visualizations, and in-depth analysis of individual concepts, identifying key factors contributing to the breakdowns and establishing a better understanding of underlying challenges. To mitigate these issues, we propose leveraging a robust optimization technique called sharpness-aware minimization (SAM). By improving the quality of intermediate concept predictions, SAM enhances both the subsequent concept-level interpretability and final target prediction performance.

## 1 INTRODUCTION

Recent advancements in deep learning have led to significant progress in a wide range of applications (LeCun et al., 2015; Brown, 2020). However, neural network models often remain as "black-boxes", making their decision-making processes challenging to interpret and control (Esteva et al., 2019; Miller, 2019). To address this, concept bottleneck models (CBMs) stand out as a promising solution, aiming to enhance model interpretability by introducing an intermediate step that relates the input and the final target prediction to the human-interpretable *concepts* (Koh et al., 2020; Bahadori & Heckerman, 2021; Sawada & Nakamura, 2022). For example, instead of relying solely on raw pixel data, CBMs can classify an animal's species based on interpretable concepts such as tail shape or body color, offering a more transparent and understandable decision-making process.

While CBMs show great promise, they come with a significant challenge: the need for labeled target and concept data during training, which requires extensive additional concept annotations. This annotation process is highly susceptible to errors; subjective interpretations of concepts, variability in annotator expertise, and simple human mistakes can all lead to mislabeled data. These issues can potentially make CBMs particularly vulnerable to noisy labels, undermining their reliability. Consequently, the very foundation of CBMs—their interpretability—can be compromised, leading to unstable target predictions and raising serious concerns about their practical effectiveness and trustworthiness.

Despite the increased susceptibility of CBMs to label noise, the impact of such noise on these models has been largely overlooked in existing research. Surprisingly, there has been no systematic study addressing how label noise affects the performance and interpretability of CBMs. For example, previous work has predominantly focused on enhancing task performance (Sawada & Nakamura, 2022; Zarlenga et al., 2022), tackling confounding issues such as information leakage (Bahadori & Heckerman, 2021; Margeloiu et al., 2021a; Mahinpei et al., 2021a), or proposing intervention methods (Chauhan et al., 2022; Shin et al., 2023), to name a few.

This paper presents the first systematic study addressing the unexplored issue of label noise in CBMs, shedding light on its detrimental effects on both model performance and interpretability. We start by investigating the extent to which label noise impacts CBMs, demonstrating that even moderate levels of noise can severely undermine their effectiveness (Section 3). To gain deeper insights, we conduct an in-depth analysis using feature visualizations and concept properties, examining how label noise disrupts the relationship between input data, intermediate concepts, and final predictions (Section 4). Last but not least, we evaluate the effectiveness of existing label noise mitigation techniques (?Baek et al.), with a primary focus on sharpness-aware minimization (SAM) (Foret et al., 2021) (Section 5). Our findings highlight the specific challenges that CBMs face under noisy conditions and provide actionable insights into building more stable and reliable interpretable models.

## 2 SETTINGS

**Concept bottleneck models.** CBMs are supervised classification models trained on a collection of an input image $x \in \mathbb{R}^d$, concepts $c \in \{0,1\}^k$, and a target $y \in \mathbb{R}$, where $d$ and $k$ denotes the dimension of input and the number of concepts, respectively. In general, CBMs operate in two stages: a concept predictor $g$ maps input images to concepts, and a target predictor $f$ uses these concepts to predict the final target (see Figure 1). Typically, $g$ is implemented as a deep neural network (*e.g.*, InceptionV3), while $f$ is a shallow neural network (*e.g.*, simple linear model). This general structure is common across various CBM variants (Zarlenga et al.,

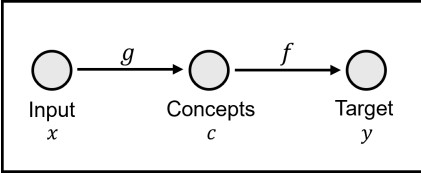

Figure 1: CBM prediction workflow. CBMs first predict an intermediate set of human-specified concepts $c$, an additional step compared to E2E models, and then use $c$ to predict the final target $y$.

2022; Yuksekgonul et al., 2023; Kim et al., 2023). A key strength of CBMs is their interpretability, as they reveal how specific concepts contribute to the final prediction. Instead of relying on raw data, CBMs make decisions through a clear combination of human-interpretable concepts, enhancing transparency.

**CBM training strategies.** To effectively train the $g$ and $f$ models within CBMs, Koh et al. (2020) introduce three different training strategies, which we also consider in our study:
- Independent(Ind): $g$ and $f$ are trained independently, with $f$ using ground-truth concepts as inputs for training.
- Sequential(Seq): $g$ is trained first, and then $f$ is trained sequentially. $f$ takes the predicted concepts as inputs from trained $g$.
- Joint(Joi): $g$ and $f$ are trained jointly at the same time as a multi-objective.

**Experimental setup.** To investigate the impact of label noise on CBM performance, we train CBMs on CUB (Wah et al., 2011) and AwA2 (Xian et al., 2018) datasets. We use InceptionV3 (Szegedy et al., 2016b) pre-trained on ImageNet (Deng et al., 2009) as a backbone for concept predictor $g$, and use a one-layer linear model for target predictor $f$, following previous standard implementations. Each experiment is repeated with three different random seeds, and we report the average performance across these runs. Detailed experimental settings are provided in Appendix G.

**Noisy dataset.** Here, we want to assess and investigate the impact of the label noise on CBMs. For investigation, first we have to build a new CBMs dataset, which can mimic the real-world noisy dataset. Thus, we define different types of noise as follows: concept noise $\widehat{c}$ refers to noise added to concept labels, while target noise $\widehat{y}$ refers to noise in target labels. Label noise encompasses both $\widehat{c}$ and $\widehat{y}$. To generate a noisy dataset, we randomly flip each label with an equal probability $\gamma$. For a dataset with $N$ classes, each incorrect label has a $1/(N-1)$ chance of being chosen. Specifically, for target noise, $\gamma\%$ of target labels are flipped, while for concept noise, $\gamma\%$ of concept labels are flipped within each target. We vary the noise rate $\gamma$ across $0\%, 10\%, 20\%, 30\%, 40\%$, where $0\%$ represents a clean dataset. This systematic approach allows us to assess how increasing levels of noise affect CBM performance. We describe more detail in Appendix A.

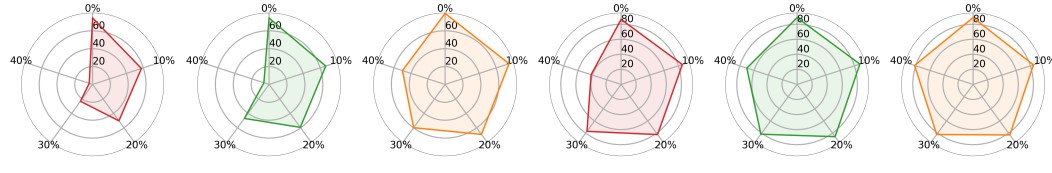

(a) CUB, `Ind`   (b) CUB, `Seq`   (c) CUB, `Joi`   (d) AWA2, `Ind`   (e) AWA2, `Seq`   (f) AWA2, `Joi`

Figure 2: Target prediction accuracy of CBMs on noisy CUB and AWA2. In the radar chart, the graphs of `Ind`, `Seq`, and `Joi`, show the performance of target prediction accuracy, demonstrating that CBMs are vulnerable to label noise across various noise levels.

Table 1: Concept prediction accuracy of CBMs on noisy CUB and AWA2. Concept accuracies below 75% are highlighted, indicating significantly reduced interpretability under label noise.

| Noise | CUB | | | | | AWA2 | | | | |
|---|---|---|---|---|---|---|---|---|---|---|
| | 0% | 10% | 20% | 30% | 40% | 0% | 10% | 20% | 30% | 40% |
| `Ind` | 96.5 | 93.8 | 91.6 | 89.1 | 85.4 | 78.5 | 78.4 | 78.1 | 77.3 | 75.3 |
| `Seq` | 96.5 | 93.8 | 91.6 | 89.1 | 85.4 | 78.5 | 78.4 | 78.1 | 77.3 | 75.3 |
| `Joi` | 92.4 | 85.9 | 78.4 | 67.6 | 57.3 | 77.8 | 74.2 | 70.1 | 65.4 | 57.4 |

## 3 CRITICAL IMPACT OF LABEL NOISE ON CBMs

Understanding the actual impact of label noise on CBMs is critical, and this section is dedicated to that exploration. To evaluate this impact, we begin by measuring model performance across various levels of label noise. By systematically analyzing the behavior of CBMs under different noise conditions, we aim to determine the extent to which label noise adversely affects their performance and interpretability. Specifically, we train CBMs on noisy versions of the CUB and AWA2 datasets, introducing noise rates ranging from 0% to 40%. We then assess the models on clean test datasets, thoroughly investigating both their final performance and the integrity of their interpretability under increasing noise levels.

**Impacts on target performance.** We start by examining the impact of label noise on CBM target performance. In Figure 2, each corner represents the target accuracy at different noise rates $\gamma$%, with larger and more regular pentagonal shapes indicating higher robustness. The results clearly show that CBMs are highly vulnerable to label noise. As the noise level increases, target performance significantly declines across all datasets. In the case of `Joi` model trained on AWA2, the performance drop might not be immediately apparent, but it still decreases by 7.2% (from 88.9% to 81.7%). The decline is even more drastic for `Ind` and `Seq` models, which almost collapse entirely at a 40% noise rates on CUB dataset. This substantial decline highlights that label noise severely compromises CBM target performance, posing a critical challenge to their reliability. We also provide the results for label noise in E2E models as a reference, which shows better resilience to label noise (see Appendix B).

**Impacts on interpretability.** Next, we assess interpretability by evaluating concept prediction accuracy, measuring how closely the predicted concept representations align with the ground-truth concept labels, as presented in Table 1. Although the `Joi` model appeared less affected by label noise in the previous section, a closer examination reveals a different story. Its concept prediction accuracy is notably worse than that of the `Ind` and `Seq` models, particularly at higher noise levels. For instance, at a 40% noise rate, the concept accuracy of the `Joi` model drops to nearly 50% across all datasets, indicating almost random predictions given that the concepts are binary. This outcome can be expected when considering that the `Joi` model learns $g$ and $f$ simultaneously, with training primary focusing on the final prediction. As a result, the concept predictor struggles more under noisy conditions. This aligns with the performance-interpretability trade-off discussed in prior work (Rudin et al., 2022). Overall, these results confirm that as label noise increases, CBMs experience severe compromises to their interpretability despite maintaining some target accuracy.

**Where is the source?** To identify the source of the detrimental effects observed earlier, we compare target performance under concept noise, target noise, and combined (*i.e.*, concept + target)

noise conditions. For evaluation, we measure the target and concept prediction accuracy of `Ind` model on the test dataset, trained on the CUB dataset across varying noise levels (see Appendix C.1 for full results).

As seen in Figure 3, the target performance under concept noise alone closely resembles the results under combined noise. Given the crucial role concepts play in CBM predictions, the disruption caused by noisy concepts strongly suggests that concept noise is the primary factor driving model failure, substantially affecting both target performance and interpretability. In the next section, we investigate deeper into how concept noise affects CBMs and the mechanisms behind these detrimental effects.

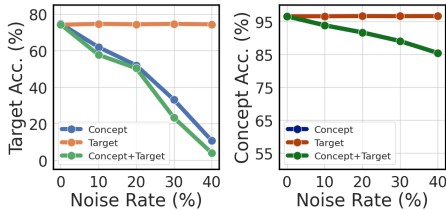

(a) Target accuracy    (b) Concept accuracy

Figure 3: Target and concept prediction accuracy of `Ind` model across noise levels.

# 4 UNDERSTANDING THE BREAKDOWN

## 4.1 CONCEPT NOISE DISRUPTS REPRESENTATION CLUSTERING

Previously, we identified that CBM failures are closely linked to concept noise. This leads to the question of how concept noise specifically disrupts the model. To answer this question, we investigate its impact on the internal representations of CBMs. Using t-SNE (Van der Maaten & Hinton, 2008), a dimensionality reduction technique, we project the final layer activations of the model to visualize how CBMs differentiate between classes under varying noise levels. This approach allows us to directly observe how concept noise affects feature clustering.

For this experiment, we select three classes: RED-WINGED BLACKBIRD, YELLOW-HEADED BLACK-BIRD, and BLACK-FOOTED ALBATROSS, and train an `Ind` CBM model on the CUB dataset with noise rates of 0%, 20%, and 40%. The first two classes are semantically similar, while the third is distinct. Ideally, in the feature space, all three classes should form separate clusters, with the similar classes clustering closer together than the distinct one. The results are shown in Figure 4.

The activation projection from the model trained on the clean dataset (*i.e.*, 0% noise) reveals well-formed, tight, and distinct clusters, with semantically similar classes close to each other (see Figure 4a). However, as concept noise is introduced, the model's ability to form clear clusters diminishes (see Figure 4b). At a 20% noise level, the clusters become broader and less distinct, with semantically similar classes starting to overlap. At 40% noise, the clusters collapse entirely, making it difficult to distinguish between the classes. This indicates that the model struggles to maintain a reliable mapping between input and target through the intermediate concepts under high concept noise.

Interestingly, when we examine the effects of target noise (see Figure 4c), the model's clusters remain tight and well-separated, regardless of the noise rate. This demonstrates that target noise does not significantly hinder the model from learning meaningful representations. Therefore, it is evident that concept noise is the primary factor that severely impairs the representation learning of CBMs.

## 4.2 CONCEPT NOISE DISTORTS CONCEPT-TARGET RELATIONSHIPS

In the previous section, our results reveal that the noise disrupts the mapping from input to target through the intermediate concepts. First, our goal is to understand how training with noisy concept data alters the relationship between concepts and their corresponding targets. To investigate this, we analyze the weight magnitudes of the $f$ model to observe how the importance of each concept changes under noise. Specifically, we plot the weight assigned to each concept for a particular target class. For this analysis, we use the `Ind` model trained on the CUB dataset under concept noise at levels of 0%, 20%, and 40%. We focus on one class, LE CONTE SPARROW, and identify the top 5 concepts with the highest weights assigned by $f$.

**Changes in concept importance.** Figure 5b shows that, in the clean dataset setting, the five most influential concepts for LE CONTE SPARROW are 'white upperparts', 'grey back', 'iridescent breast', 'yellow upperparts', and 'yellow upper tail', indicating that $f$ heavily relies on these concepts for

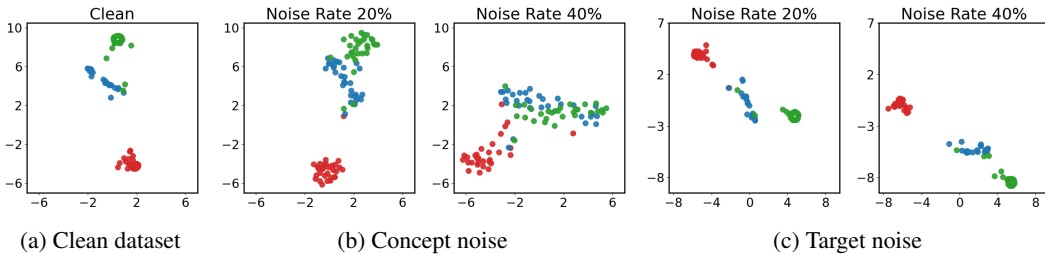

(a) Clean dataset  (b) Concept noise  (c) Target noise

Figure 4: t-SNE visualization of the activations at the last layer in Ind model. We visualize three classes: BLACK-FOOTED ALBATROSS (●), RED-WINGED BLACKBIRD (●), and YELLOW-HEADED BLACKBIRD (●). The t-SNE plots show models trained on (a) a clean dataset, (b) concept noise, and (c) target noise, highlighting how noise affects feature representation and class separation.

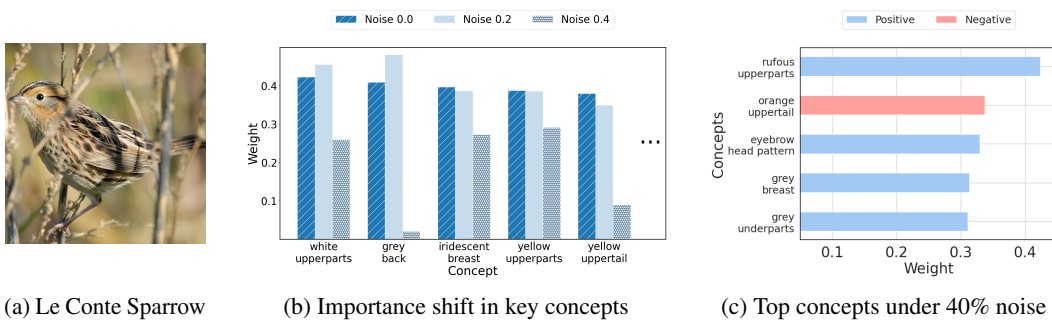

(a) Le Conte Sparrow  (b) Importance shift in key concepts  (c) Top concepts under 40% noise

Figure 5: Collapse in concept relationship. (b) shows top 5 most influential concepts for LE CONTE SPARROW in a clean setting and tracks how their influence shifts as noise increases. (c) presents top 5 influential concepts under 40% noise. In both figures, bars in blue denote positive weights, while red indicate negative weights, illustrating how noise alters concept relevance and disrupts interpretability.

target prediction. As noise increases, however, these relationships shift drastically, with many of these concepts losing their importance. For example, while 'grey back' is initially the second most important, it overtakes 'white upperparts' at 20% noise but then sharply drops in significance at 40%. This reveals that concept noise disrupts the model's ability to maintain a stable connection between the target and its key concepts.

**Building an incorrect relationship.** At a 40% noise level, we observe that the ranking of important concepts changes entirely compared to the clean dataset (see Figure 5c), indicating that the relationships between target and concepts are fundamentally altered by concept noise. Furthermore, we observed the concept 'orange uppertail' emerges but has a negative weight. Here, the negative weights indicate that the presence of such concepts lowers the probability of predicting the corresponding target. This suggests that the model fails to associate this concept 'orange uppertail' with LE CONTE SPARROW, resulting in the negative influence on making correct task predictions. These findings highlight that concept noise causes CBMs to build an incorrect relationship with concepts to targets, resulting in a deterioration of the model's predictive accuracy.

Our next objective is to investigate how the relationships between the input and its individual concept are affected by concept noise, thereby providing insights into how $g$'s output influences $f$ during evaluation. We assess this by examining the accuracy of each concept predicted by $g$. For this, we used the same Ind model trained on the CUB dataset under concept noise levels of 0%, 20%, and 40%. We focused on the concept prediction accuracy for the single class, LE CONTE SPARROW.

**Inconsistent individual concept accuracy.** As shown in Figure 6, the accuracy differences among concepts in a clean setting are not severe. However, as concept noise increases, these differences become much more pronounced, and the accuracy drops for each concept become highly uneven. For instance, even though all concepts are exposed to a similar level of noise, some concepts experience a dramatic decline in accuracy compared to others, reflecting that the impact of noise varies significantly across concepts.

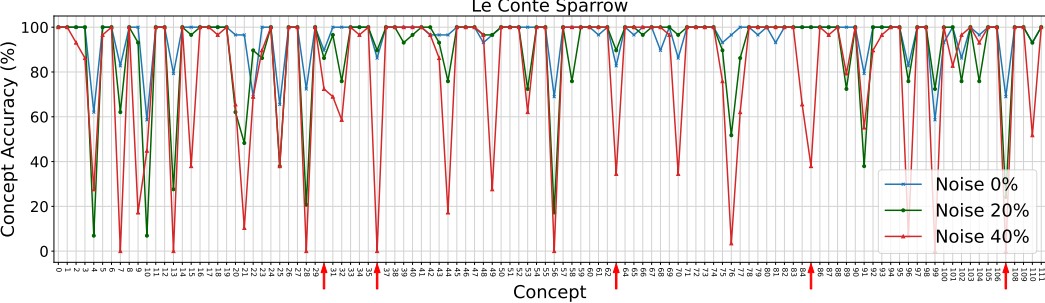

Figure 6: Impact of noise on individual concept prediction accuracy. We evaluate how noise affects individual concepts for the LE CONTE SPARROW. As noise levels increase, the accuracy drops across concepts become highly uneven, and leading to incorrect target predictions.

**Impact on target predictions.** Critically, when a concept that plays a major role in target prediction suffers a sharp drop in accuracy, it greatly hinders $f$'s ability to make correct predictions. For example, in the case of LE CONTE SPARROW, the top five critical concepts from the clean dataset, highlighted in red in Figure 6, show significant accuracy declines under noisy conditions. This means that $f$ ends up relying on inaccurate or less relevant concepts, leading to incorrect target predictions. Thus, concept noise not only disrupts individual concept accuracy but also severely impacts overall target prediction accuracy in CBMs.

## 5 MITIGATING THE NOISE EFFECTS IN CBMS

In this section, we address how to mitigate the detrimental effects of label noise in CBMs, with a focus on sharpness-aware minimization (SAM) (Foret et al., 2021). We examine the impact of SAM on improving both target prediction accuracy and model interpretability under noisy conditions. We begin by providing the background of SAM and how SAM effectively manages noisy dataset training (Section 5.1). Next, we demonstrate how integrating SAM significantly enhances CBM robustness in noisy settings, supported by an analysis of the underlying reasons (Section 5.2). Further mitigation techniques are discussed in Appendix F.

### 5.1 BACKGROUND: SHARPNESS-AWARE MINIMIZATION

The geometry of the loss landscape is closely linked to a generalization ability of model, with flatter minima often resulting in better generalization performance, as demonstrated by several studies (Dinh et al., 2017; Li et al., 2018). Building on this, Foret et al. (2021) introduced SAM, which targets minimizing the sharpness of the loss landscape to achieve flatter minima. Notably, SAM has proven effective not only for enhancing generalization but also in managing noisy label settings (Baek et al.), as it encourages the model to prioritize learning from clean data over fitting to noisy labels.

### 5.2 SAM IMPROVES ROBUSTNESS OF CBMS

In this section, we investigate how using SAM affects CBM performance under different noise conditions and whether it effectively improves robustness. For this, we trained CBMs on the CUB and AwA2 datasets under concept, target, and combined noise settings with noise rates of 0%, 20%, and 40% across all training strategies. The complete results can be found in Appendix D.1.

Table 2 presents the target and concept prediction performance of SAM under the combined noise setting. For comparison, we include results from the baseline, *i.e.*, CBM trained with the standard SGD optimizer. The results indicate that SAM consistently outperforms SGD across almost all noise settings, significantly enhancing both target and concept prediction performance. On average, SAM achieves gains of 0.6%, 0.6%, and 0.9% in concept prediction accuracy, and 3.2%, 2.8%, and 2.4% in target prediction accuracy for the `Ind`, `Seq`, and `Joi` models, respectively. Interestingly, even a modest improvement in concept prediction accuracy leads to substantial gains in target prediction accuracy. For example, in the `Ind` model trained on the AwA2 dataset with a 20% noise ratio, a mere 0.4% improvement in concept prediction accuracy results in a 3.4% boost in target prediction

Table 2: Comparison of test accuracy between SGD and SAM. CBMs trained with the SAM optimizer on CUB and AWA2 datasets under combined noise conditions demonstrate significantly more robust performance than those trained with SGD. The noise rate is indicated by $nr$.

| CBM Type | Optimizer | Metric | CUB | | | AWA2 | | | $\Delta$ |
| --- | --- | --- | --- | --- | --- | --- | --- | --- | --- |
| | | | $nr = 0\%$ | $nr = 20\%$ | $nr = 40\%$ | $nr = 0\%$ | $nr = 20\%$ | $nr = 40\%$ | |
| Ind | SGD | concept acc | $96.5_{\pm 0.0}$ | $91.6_{\pm 0.0}$ | $85.4_{\pm 0.0}$ | $78.5_{\pm 0.8}$ | $78.1_{\pm 0.6}$ | $75.3_{\pm 0.5}$ | |
| | | target acc | $74.3_{\pm 0.3}$ | $50.3_{\pm 0.7}$ | $4.0_{\pm 0.7}$ | $86.5_{\pm 0.8}$ | $82.3_{\pm 1.1}$ | $41.9_{\pm 1.0}$ | |
| | SAM | concept acc | $97.2_{\pm 0.1}$ | $92.5_{\pm 0.1}$ | $86.3_{\pm 0.1}$ | $78.8_{\pm 0.7}$ | $78.5_{\pm 0.6}$ | $75.8_{\pm 1.2}$ | **+0.6** |
| | | target acc | $79.0_{\pm 0.8}$ | $54.2_{\pm 0.7}$ | $5.0_{\pm 1.4}$ | $87.8_{\pm 0.7}$ | $85.7_{\pm 0.4}$ | $46.5_{\pm 1.6}$ | **+3.2** |
| Seq | SGD | concept acc | $96.5_{\pm 0.0}$ | $91.6_{\pm 0.0}$ | $85.4_{\pm 0.1}$ | $78.5_{\pm 0.8}$ | $78.1_{\pm 0.7}$ | $75.3_{\pm 0.8}$ | |
| | | target acc | $74.2_{\pm 0.2}$ | $59.3_{\pm 0.6}$ | $6.1_{\pm 2.6}$ | $88.7_{\pm 0.2}$ | $85.8_{\pm 0.3}$ | $70.1_{\pm 3.9}$ | |
| | SAM | concept acc | $97.2_{\pm 0.1}$ | $92.5_{\pm 0.1}$ | $86.3_{\pm 0.1}$ | $78.8_{\pm 0.8}$ | $78.5_{\pm 0.5}$ | $75.9_{\pm 1.3}$ | **+0.6** |
| | | target acc | $78.4_{\pm 0.5}$ | $63.5_{\pm 0.9}$ | $10.7_{\pm 6.0}$ | $90.5_{\pm 0.4}$ | $88.0_{\pm 0.5}$ | $69.6_{\pm 6.3}$ | **+2.8** |
| Joi | SGD | concept acc | $91.9_{\pm 0.7}$ | $78.4_{\pm 0.6}$ | $57.3_{\pm 0.3}$ | $77.8_{\pm 0.5}$ | $70.1_{\pm 0.8}$ | $57.4_{\pm 0.2}$ | |
| | | target acc | $81.4_{\pm 0.1}$ | $69.2_{\pm 0.5}$ | $50.1_{\pm 0.5}$ | $88.9_{\pm 0.1}$ | $83.0_{\pm 0.3}$ | $81.7_{\pm 0.3}$ | |
| | SAM | concept acc | $92.2_{\pm 0.5}$ | $78.5_{\pm 0.1}$ | $57.9_{\pm 0.3}$ | $78.0_{\pm 0.4}$ | $72.7_{\pm 0.4}$ | $58.9_{\pm 0.9}$ | **+0.9** |
| | | target acc | $81.4_{\pm 0.6}$ | $69.9_{\pm 0.6}$ | $50.6_{\pm 1.5}$ | $91.9_{\pm 0.3}$ | $88.4_{\pm 0.2}$ | $86.6_{\pm 0.3}$ | **+2.4** |

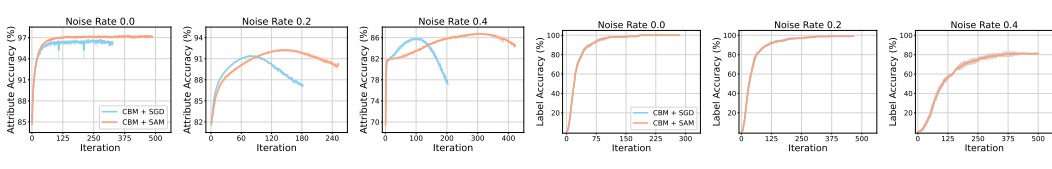

(a) Concept validation accuracy.         (b) Target validation accuracy.

Figure 7: Training progress of Ind under combined noise condition. The blue line represents the model trained with SGD, while the orange line indicates training with SAM.

accuracy. This aligns with our earlier findings that accurately predicted concepts are critical in CBMs, as they play a direct role in predicting a reliable target.

In the noise setting, the effectiveness of SAM is mainly driven by finding flatter minima in the loss landscape, which reduces sensitivity to noisy labels and allows the model to focus on learning cleaner concept representations. By doing so, SAM ensures that even under noisy conditions, CBMs maintain more reliable and robust connections between concepts and target predictions, resulting in overall enhanced performance.

To demonstrate SAM's impact on improving concept prediction reliability during evaluation, we analyzed the performance of individual concepts (see Figure 17 in Appendix E). We observe that SAM improves accuracy across nearly all concepts, particularly in noisy settings. Notably, key concepts critical for target prediction, such as those for classifying LAYSAN ALBATROSS—eyeline (102), brown wing (10), hooked seabird-shaped bill (4), brown upperparts (25), and dagger-shaped bill (1)—show substantial accuracy gains with SAM. These improvements may contribute to more accurate target predictions, as SAM helps the model learn more reliable and distinct concept representations, even under noisy settings.

Figure 7 compares the training progress of SAM and SGD, showing that SAM trains more accurate concepts under noisy label settings (see Appendix D.2 for full results). While the target prediction model performs similarly with both SAM and SGD, SAM significantly outperforms SGD in training the concept prediction model. SGD initially learns faster but tends to overfit to noise, resulting in poorer validation accuracy over time. In contrast, SAM effectively mitigates overfitting and achieves better validation performance for concept predictions. This indicates that even when the target prediction model captures the concept-to-target relationship equally well, the reliability of concepts predicted by model trained on SAM leads to substantial improvements in overall target accuracy. These findings further validate our earlier insights, emphasizing that accurate concept predictions are crucial for CBM performance. SAM's ability to generate clearer concept representations directly enhances target accuracy, while SGD's vulnerability to noise undermines model reliability.

Table 3: Impact of label noise on different architectures. CBMs with ResNet-18 and ViT-B/16 exhibit significant vulnerability to label noise, but SAM effectively mitigates this performance drop across noise settings.

| Backbone | Noise Loc | Metric | SGD | | | SAM | | | Δ |
|---|---|---|---|---|---|---|---|---|---|
| | | | $nr = 0\%$ | $nr = 20\%$ | $nr = 40\%$ | $nr = 0\%$ | $nr = 20\%$ | $nr = 40\%$ | |
| ResNet-18 | Combined | concept acc | 95.23 | 90.40 | 81.28 | 95.98 | 92.70 | 81.78 | **+1.19** |
| | | target acc | 69.14 | 49.14 | 0.90 | 73.32 | 60.55 | 0.52 | **+5.07** |
| ViT-B/16 | Combined | concept acc | 96.04 | 89.06 | 82.76 | 96.74 | 90.95 | 85.84 | **+1.89** |
| | | target acc | 73.66 | 31.05 | 1.69 | 77.87 | 47.26 | 3.19 | **+7.31** |

# 6 FURTHER ANALYSIS

## 6.1 CBM VARIANTS

We examine how label noise affects CBM variants, specifically Concept Embedding Models (CEMs) (Zarlenga et al., 2022) and Energy-based Concept Bottleneck Models (ECBMs) (Xu et al., 2024). CEMs use positive and negative embeddings to capture meaningful concepts, while ECBMs employ a joint energy model encompassing input, concepts, and the target. Both models are trained under combined noise at varying level on the CUB dataset, following their original training protocols.

Figure 8 displays the results, revealing both models are vulnerable to label noise, confirming that concept-based models are significantly affected by label noise. Although these variants aim to enhance interpretability and target classification, they struggle to maintain robustness against label noise, indicating that even advanced concept-based models remain susceptible to noise disruption. These results emphasize the critical need to further investigate the effects of label noise on concept-based models.

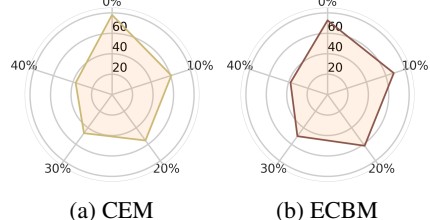

(a) CEM   (b) ECBM

Figure 8: Target prediction accuracy of CEM and ECBM on noisy CUB.

## 6.2 OTHER TYPES OF NOISE

In real-world settings, label noise often stems from ambiguous or systematically mislabeled data, which can significantly degrade model performance. To evaluate the impact of more practical label noise on CBMs, we introduce pairwise noise, where label $i$ flips to $i + 1$ (mod $N$), forming a structured cyclical pattern of label corruption. This simulates a more realistic, non-random label noise scenario compared to symmetric noise. We trained CBMs on the CUB dataset across varying noise levels using this pairwise noise.

Figure 9 shows the results for the Ind model trained under different noisy conditions. The findings reveal that CBMs are also vulnerable to pairwise noise, exhibiting significant performance drops, particularly under combined and concept noise settings. This aligns with the earlier symmetric noise results, but with even lower performance. Notably, under class noise, the performance of model deteriorates sharply, when noise reaches 40%. This suggests that the structured noise hinders the model's ability to learn true label distributions, making pairwise noise more detrimental than random symmetric noise. These results emphasize the need for effective strategies to handle label noise in CBMs (see Appendix C.2 for full results).

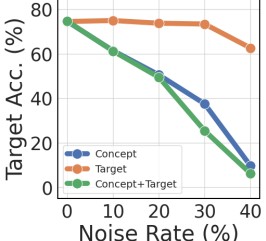

Figure 9: Target prediction accuracy on pairwise noise.

## 6.3 DIFFERENT ARCHITECTURES

Given that each backbone architecture possesses distinct training capabilities, their sensitivity to label noise in CBMs may vary. To investigate this, we evaluated the impact of label noise on CBMs trained with convolutional networks, such as ResNet-18 (He et al., 2016), and transformer networks, such as ViT-B/16 (Dosovitskiy et al., 2021), across noisy CUB datasets with noise ratios of 0%, 20%, and 40%. Furthermore, we compared the performance of the SAM optimizer with SGD across these architectures to evaluate its effectiveness.

Table 3 shows the target and concept prediction accuracy of `Ind` models trained with different backbones using both SGD and SAM under combined noise settings. Despite the architectural differences, all CBMs struggled to maintain their performance under label noise, consistent with the degradation observed in InceptionV3. Notably, SAM consistently mitigates performance drops across all architectures, yielding average improvements of 5.07% for ResNet-18 and 7.31% for ViT-B/16 compared to SGD. These results indicate that while the choice of backbone architecture has a relatively minor impact on noise robustness, the SAM optimizer plays a crucial role in enhancing resilience to label noise, suggesting it as a promising training strategy for CBMs in noisy environments.

## 7 RELATED WORKS

**Concept bottleneck models.** Koh et al. (2020) introduced Concept Bottleneck Models (CBMs), which generate a "bottleneck concept" to predict the final target based on that concept, which improves the interpretability of standard end-to-end models. Building on this foundational work, CBMs have been studied and further developed in various ways. These include improving task performance (Zarlenga et al., 2022; Yuksekgonul et al., 2023; Kim et al., 2023; Xu et al., 2024), enhancing intervention capabilities (Xu et al., 2024; Chauhan et al., 2022; Sheth et al., 2022; Shin et al., 2023), and improving interpretability (Mahinpei et al., 2021b; Margeloiu et al., 2021b; Marconato et al., 2022) in supervised learning settings. Despite these advancements, most prior work has primarily focused on studying CBMs in clean, noise-free datasets, limiting their applicability to real-world conditions where data is often noisy or mislabeled. In this work, we aim to address this limitation by investigating the impact of label noise on CBMs across various scenarios, and further analyzing their influence on various aspects.

**Noisy label learning.** Since noisy labels can significantly impair the generalization ability of deep neural networks, developing robust training techniques to handle noisy data has become a crucial challenge in modern deep learning applications. To address this issue, various approaches have been proposed to mitigate the detrimental effects of label noise. Earlier approaches primarily focus on adjusting the loss function to mitigate the effects of noise. One strategy involves modifying the loss by applying an estimated noise transition matrix (Patrini et al., 2017; Hendrycks et al., 2018; Xia et al., 2019; Yao et al., 2020), while others re-weight the loss to help deep neural networks focus on correctly labeled samples (Liu & Tao, 2015). Robust loss functions (Natarajan et al., 2013; Ghosh et al., 2017; Zhang & Sabuncu, 2018; Wang et al., 2019; Amid et al., 2019; Liu & Guo, 2020), robust regularizers (Liu et al., 2020; Xia et al., 2020; Cheng et al., 2021), and robust optimizer (Baek et al.; Tanaka et al., 2018) have also been studied to handle label noise effectively.

## 8 CONCLUSION

**Conclusion.** The impact of label noise on CBMs is critical yet previously underexplored, particularly concerning how it affects interpretability and reliability in real-world applications. Our comprehensive study reveals that CBMs are highly sensitive to label noise, with concept label noise being a primary factor that significantly impairs both target prediction accuracy and interpretability. We demonstrated that this noise undermines the representation learning, and find that it disrupts not only the concept-target relationship, but also the input-concept relationship, leading to degraded model performance. By incorporating the SAM optimizer, we effectively mitigated these detrimental effects, enhancing both concept prediction and target accuracy across varying noise levels. Our findings emphasize the need for noise-aware training strategies in CBMs to maintain their interpretability and reliability, suggesting SAM as a promising solution.

**Limitations and future works.** While our study offers a comprehensive analysis of the impact of label noise on CBMs, there are several limitations: (i) Our work primarily focuses on analyzing the effects of label noise, with the exploration of mitigation techniques being less extensive. Although we show that SAM effectively mitigates some negative effects, further investigation into alternative optimization methods or training strategies remains an open avenue for future research. (ii) We restricted our experiments to certain datasets (*e.g.*, CUB and AwA2) and architectures (*e.g.*, InceptionV3, ResNet-18, ViT-B/16). Future work could explore other datasets and more diverse architectures to understand how label noise impacts CBMs in various settings. Addressing these limitations could further advance our understanding and robustness of CBMs under noisy conditions.

## REPRODUCIBILITY STATEMENT

To ensure the reproducibility of our findings, we provide a comprehensive description of our experimental setup and evaluation procedures in Appendix G. This study utilizes publicly available datasets, and detailed preprocessing instructions are also included. Upon publication, the source code, along with implementation details and hyperparameter configurations, will be made available in a public repository. Furthermore, we will specify all software dependencies, version details, and hardware configurations used in our experiments to facilitate accurate reproduction of our results.

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

CONTENTS

## A  LABEL NOISE SETTING

We explore two types of synthetic label noise: symmetric noise and pairwise noise, as illustrated in Figure 10. These noise models are commonly employed in existing literature (Ma et al., 2018; Thulasidasan et al., 2019; Pleiss et al., 2020; Wang et al., 2021; Gui et al., 2021). The noise settings are defined as follows:

1. **Symmetric noise**: This noise type is introduced by randomly flipping labels in each class to any other class label with equal probability. For example, in a dataset with $N$ classes, each label has a $\frac{1}{N-1}$ chance of being incorrectly reassigned to any of the remaining $N-1$ classes (See Figure 10a).

2. **Pairwise noise**: This noise model involves flipping each label to its adjacent class label. For instance, if the classes are ordered sequentially from 1 to $N$, a label $i$ will be flipped to $i+1 \pmod{N}$, creating a cyclical pattern (See Figure 10b).

(a) Symmetric Noise.     (b) Pairwise Noise.

Figure 10: Two types of synthetic label noises.

The symmetric noise setting simulates a scenario where annotation errors are uniformly distributed across all classes, representing general labeling uncertainty. In contrast, the pairwise noise setting reflects situations where labels are systematically confused with their nearest counterparts, a common occurrence in tasks involving ordinal data or closely related categories. In previous studies, a dataset was created through majority voting and assumed to be the true dataset. Following this approach, we inject noisy labels into the majority-voted dataset to generate the noisy dataset. Specifically, to create the concept noisy data, for a given sample $x_i, c_i, y_i$, we alter the concept labels by $\gamma\%$ within the concept set $c_i$ under the concept noise setting. For target noise, we modify the class label by $\gamma\%$ across the entire dataset. By integrating these noise models, we aim to evaluate the robustness of CBMs under varying types of label corruption.

## B  COMPARISON CBMS BETWEEN E2E MODELS

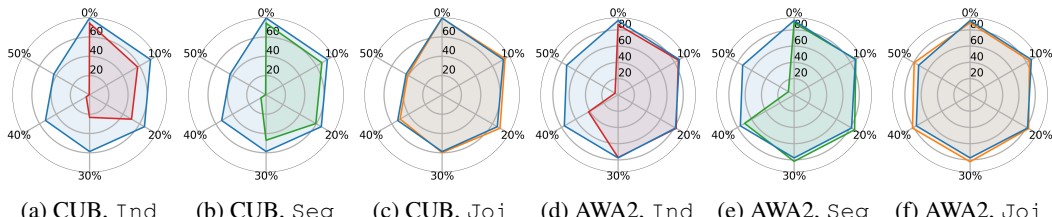

(a) CUB, Ind  (b) CUB, Seq  (c) CUB, Joi  (d) AWA2, Ind  (e) AWA2, Seq  (f) AWA2, Joi

Figure 11: Target prediction accruacy of CBMs and E2E on noisy CUB and AWA2. In the radar chart, the graphs of Ind, Seq, Joi, and E2E, showing the performance of target prediction accuracy.

Figure 11 represents target and concept prediction accuracy under different noise settings compared with the End-to-End model. E2E models demonstrate greater resilience to noise, due to their over-parameterized nature, which allows them to fit noisy data more effectively (Zhang et al., 2021; Liu et al., 2022). We provide these results to demonstrate that such breakdowns do not always occur, contrary to what is often observed in CBMs. Here, we note that the direct comparison between CBMs and E2E models is not fair.

## C  IMPACT OF DIFFERENT NOISE TYPE ON CBM

### C.1  RESULTS ON SYMMETRIC NOISE SETTING

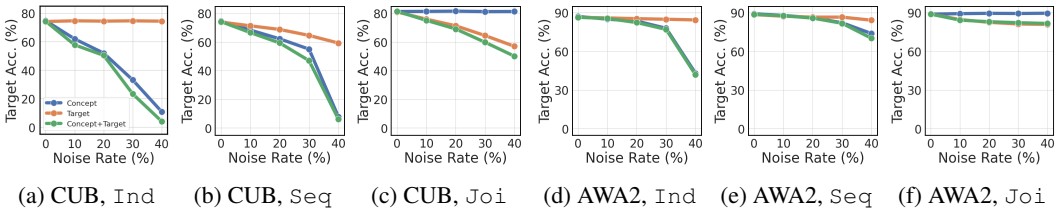

(a) CUB, Ind  (b) CUB, Seq  (c) CUB, Joi  (d) AWA2, Ind  (e) AWA2, Seq  (f) AWA2, Joi

Figure 12: Target prediction accuracy of CBMs on noisy concept/target/combined CUB and AWA2. These results offer insight into how different noise types affect target prediction performance.

Table 4: Concept prediction accuracy of CBMs on noisy concept/target/combined CUB and AWA2. Concept accuracies below 75% are highlighted, indicating significantly reduced interpretability in the presence of label noise.

| | CUB | | | | | | | | | AWA2 | | | | | | | | |
| | Concept | | | Class | | | Combined | | | Concept | | | Class | | | Combined | | |
| Noise | Ind | Seq | Joi | Ind | Seq | Joi | Ind | Seq | Joi | Ind | Seq | Joi | Ind | Seq | Joi | Ind | Seq | Joi |
|---|---|---|---|---|---|---|---|---|---|---|---|---|---|---|---|---|---|---|
| 0% | 96.6 | 96.6 | 92.4 | 96.5 | 96.5 | 91.9 | 96.5 | 96.5 | 92.4 | 78.6 | 78.6 | 77.9 | 78.5 | 78.5 | 77.7 | 78.5 | 78.5 | 77.8 |
| 10% | 93.8 | 93.8 | 87.9 | 96.6 | 96.6 | 90.3 | 93.8 | 93.8 | 85.9 | 78.2 | 78.2 | 77.4 | 78.3 | 78.3 | 77.7 | 78.4 | 78.4 | 74.2 |
| 20% | 91.7 | 91.7 | 82.2 | 96.6 | 96.6 | 89.2 | 91.6 | 91.6 | 78.4 | 78.1 | 78.1 | 77.4 | 78.3 | 78.3 | 74.2 | 78.1 | 78.1 | 70.1 |
| 30% | 89.0 | 89.0 | 71.5 | 96.6 | 96.6 | 88.4 | 89.1 | 89.1 | 67.6 | 77.3 | 77.3 | 76.8 | 78.5 | 78.5 | 74.1 | 77.3 | 77.3 | 65.4 |
| 40% | 85.4 | 85.4 | 60.3 | 96.6 | 96.6 | 87.0 | 85.4 | 85.4 | 57.3 | 75.2 | 75.2 | 73.9 | 78.4 | 78.4 | 73.2 | 75.3 | 75.3 | 57.4 |

We further examine the influence of different types of noise on CBMs and analyze how concept and target noise affect the performance of CBMs. Figure 12 shows the final target accuracy of CBMs on the CUB and AWA2 datasets, and Table 4 represents the concept prediction accuracy under different noise types.

When target noise is injected, all CBMs show a slight performance decrease in concept and final target accuracy compared to the results under combined noise. In detail, Ind maintains its performance and even improves its results in some cases while maintaining concept accuracy. Although Seq experiences some performance degradation as noise increases, it retains its concept and class accuracy, demonstrating different behavior with the combined noise setting. Joi also shows some performance drops in class accuracy.

In contrast, under concept noise, CBMs exhibit substantial vulnerability, similar to the results observed on the combined noisy dataset. Specifically, for models trained separately, such as `Ind` and `Seq`, class performance drops significantly as noise increases, eventually collapsing at high noise levels on the CUB dataset. Additionally, `Ind` degrades more rapidly than `Seq` in general. Even though the `Joi` model shows better resilience to noise in terms of class accuracy, especially on the CUB dataset, its concept prediction accuracy deteriorates faster than that of `Ind` and `Seq`, indicating that incorrect concepts are being used to predict classes.

We hypothesize that the $f$ model trained with the `Ind` type learns the relationship between noisy concepts and the final target labels, however, during the evaluation, $f$ receives concept predictions $\widetilde{c}$ from $g$, which differ from $\widehat{c}$, leading to inaccurate predictions and eventual collapse at high noise levels. While the `Seq` model learns the relationship between the predicted concepts from the $g$ model and the final target labels, allowing it to maintain better performance than the `Ind` model, it still collapses when concept errors become too large. Since `Joi` type trains both models jointly, even if it mispredicts the concepts, it can still achieve good performance by relying on patterns in the data that directly lead to correct class predictions. This leads the `Joi` type model to achieve higher target accuracy than the other types but has poor concept accuracy. Overall, these results suggest that concept noise introduces a trade-off between interpretability and final performance, ultimately compromising the performance of CBMs.

### C.2 Results on pairwise noise setting

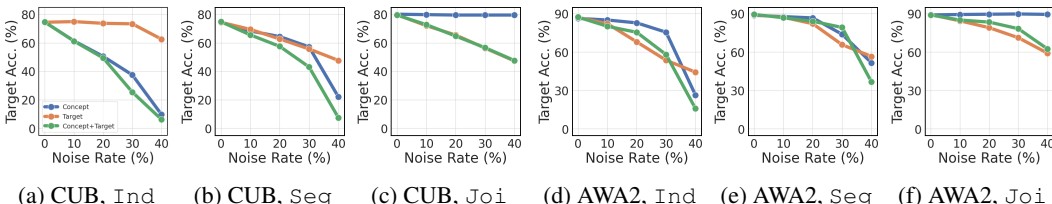

(a) CUB, `Ind`  (b) CUB, `Seq`  (c) CUB, `Joi`  (d) AWA2, `Ind`  (e) AWA2, `Seq`  (f) AWA2, `Joi`

Figure 13: Target prediction accuracy in CBMs on concept/target/combined pairwise noisy CUB. These results offer insight into how different noise types affect target prediction performance.

We trained CBMs on the CUB dataset under various noise types, using pairwise noise settings across `Ind`, `Seq`, and `Joi` models as shown in Figure 13. The overall results support our message that CBMs are highly vulnerable to label noise, which causes significant performance degradation and also causes the model collapse. We also observe similar behavior under symmetric noise, further indicating that concept noise has a substantial impact on final target performance and concept prediction accuracy. Here, we note that in some cases, pairwise noise led to a further decline in accuracy, *i.e.*, in the `Ind` and `Seq` models under target noise. These results suggest that it is more challenging for CBMs to handle the pairwise noise compared to the symmetric noise type.

## D Comparison of accuracy: SGD and SAM

### D.1 Comparison of final prediction accuracy between SGD and SAM

In Table 5, we evaluate the overall target and concept prediction accuracy of CBMs trained with SAM and SGD on the CUB dataset across various noise types, with noise rates ranging from 0% to 40%. Across all noise rates, SAM consistently outperforms SGD in both concept and target accuracy under various noise types, *i.e.*, concept, target, and label noise. These results suggest that SAM is more effective at handling label noise than SGD, maintaining higher accuracy levels across different training types. Notably, SAM proves highly effective in both `Ind` and `Seq` models, showing average performance gains across different noise levels of 0.7% and 3.9% for `Ind`, and 0.7% and 3.7% for `Seq` in concept and target accuracy, respectively.

We also evaluate the concept and final target accuracy of CBMs trained with SAM and SGD on the AWA2 dataset, as presented in Table 6. We find that even on the AWA2 dataset, SAM further enhances performance, demonstrating that SAM can mitigate the detrimental effects on CBMs regardless of the dataset. On the AWA2 dataset, SAM shows better improvement in the `Joi` type model, which is

slightly different from previous results, where the average performance gains across different noise levels are 0.2% and 4.0% in concept and target accuracy for `Joi` type, respectively.

Table 5: Comparison of test accuracy between SGD and SAM on the CUB dataset.

| Noise Loc | Optimizer | Metric | $nr = 0\%$ | $nr = 10\%$ | $nr = 20\%$ | $nr = 30\%$ | $nr = 40\%$ | $\Delta$ |
|---|---|---|---|---|---|---|---|---|
| | | | | | Noise Rate | | | |
| *Independent Bottleneck* | | | | | | | | |
| Concept (X → C) | SGD | concept acc | $96.6\pm0.1$ | $93.8\pm0.0$ | $91.7\pm0.1$ | $89.0\pm0.1$ | $85.4\pm0.1$ | |
| | | target acc | $74.7\pm0.8$ | $61.9\pm1.7$ | $52.0\pm2.1$ | $33.1\pm1.7$ | $10.8\pm2.1$ | |
| | SAM | concept acc | $97.2\pm0.1$ | $94.6\pm0.1$ | $92.5\pm0.0$ | $89.7\pm0.1$ | $86.3\pm0.0$ | **+0.8** |
| | | target acc | $78.9\pm1.0$ | $65.7\pm1.6$ | $56.9\pm1.4$ | $38.1\pm2.7$ | $11.0\pm0.2$ | **+3.6** |
| Target (C → Y) | SGD | concept acc | $96.5\pm0.0$ | $96.6\pm0.1$ | $96.6\pm0.0$ | $96.6\pm0.0$ | $96.6\pm0.0$ | |
| | | target acc | $74.2\pm0.1$ | $74.6\pm0.3$ | $74.4\pm0.2$ | $74.6\pm0.3$ | $74.4\pm0.0$ | |
| | SAM | concept acc | $97.2\pm0.0$ | $97.2\pm0.1$ | $97.2\pm0.0$ | $97.2\pm0.0$ | $97.2\pm0.0$ | **+0.6** |
| | | target acc | $78.7\pm0.5$ | $78.9\pm0.2$ | $78.4\pm0.3$ | $78.3\pm0.5$ | $79.0\pm0.3$ | **+4.2** |
| Combined (X → C, C → Y) | SGD | concept acc | $96.5\pm0.0$ | $93.8\pm0.1$ | $91.6\pm0.0$ | $89.1\pm0.0$ | $85.4\pm0.0$ | |
| | | target acc | $74.3\pm0.3$ | $57.7\pm2.0$ | $50.3\pm0.7$ | $23.3\pm1.2$ | $4.0\pm0.7$ | |
| | SAM | concept acc | $97.2\pm0.1$ | $94.6\pm0.1$ | $92.5\pm0.1$ | $89.7\pm0.1$ | $86.3\pm0.1$ | **+0.6** |
| | | target acc | $79.0\pm0.8$ | $61.8\pm1.8$ | $54.2\pm0.7$ | $28.5\pm1.4$ | $5.0\pm1.4$ | **+3.8** |
| *Sequential Bottleneck* | | | | | | | | |
| Concept (X → C) | SGD | concept acc | $96.6\pm0.1$ | $93.8\pm0.0$ | $91.7\pm0.1$ | $89.0\pm0.1$ | $85.4\pm0.1$ | |
| | | target acc | $74.6\pm0.4$ | $68.4\pm0.2$ | $62.2\pm0.5$ | $55.0\pm0.3$ | $14.7\pm12.4$ | |
| | SAM | concept acc | $97.2\pm0.1$ | $94.6\pm0.1$ | $92.5\pm0.0$ | $89.7\pm0.1$ | $86.3\pm0.0$ | **+0.8** |
| | | target acc | $78.7\pm0.4$ | $72.0\pm0.2$ | $66.9\pm0.5$ | $57.7\pm1.3$ | $17.4\pm6.1$ | **+3.6** |
| Target (C → Y) | SGD | concept acc | $96.5\pm0.0$ | $96.6\pm0.1$ | $96.6\pm0.0$ | $96.6\pm0.0$ | $96.6\pm0.0$ | |
| | | target acc | $74.0\pm0.6$ | $71.3\pm1.0$ | $68.8\pm0.6$ | $64.6\pm1.0$ | $59.3\pm2.5$ | |
| | SAM | concept acc | $97.2\pm0.0$ | $97.2\pm0.1$ | $97.2\pm0.0$ | $97.2\pm0.0$ | $97.2\pm0.0$ | **+0.6** |
| | | target acc | $77.9\pm0.3$ | $75.1\pm0.2$ | $71.3\pm1.1$ | $66.7\pm1.0$ | $63.3\pm0.9$ | **+3.6** |
| Combined (X → C, C → Y) | SGD | concept acc | $96.5\pm0.0$ | $93.8\pm0.0$ | $91.6\pm0.0$ | $89.1\pm0.0$ | $85.4\pm0.1$ | |
| | | target acc | $74.2\pm0.2$ | $66.6\pm0.4$ | $59.3\pm0.6$ | $47.0\pm1.7$ | $6.1\pm2.6$ | |
| | SAM | concept acc | $97.2\pm0.1$ | $94.6\pm0.1$ | $92.5\pm0.1$ | $89.7\pm0.1$ | $86.3\pm0.1$ | **+0.6** |
| | | target acc | $78.4\pm0.5$ | $70.5\pm0.6$ | $63.5\pm0.9$ | $50.1\pm1.1$ | $10.7\pm6.0$ | **+4.0** |
| *Joint Bottleneck* | | | | | | | | |
| Concept (X → C) | SGD | concept acc | $92.4\pm0.5$ | $87.9\pm0.1$ | $82.2\pm0.5$ | $71.5\pm0.1$ | $60.3\pm0.3$ | |
| | | target acc | $81.3\pm0.2$ | $81.4\pm0.2$ | $81.6\pm0.2$ | $81.3\pm0.2$ | $81.4\pm0.0$ | |
| | SAM | concept acc | $92.2\pm0.3$ | $87.8\pm0.5$ | $82.0\pm0.4$ | $71.0\pm0.7$ | $60.3\pm0.5$ | **-0.2** |
| | | target acc | $81.4\pm0.4$ | $81.4\pm0.4$ | $81.9\pm0.2$ | $81.4\pm0.3$ | $81.6\pm0.4$ | **+0.2** |
| Target (C → Y) | SGD | concept acc | $91.9\pm0.7$ | $90.3\pm0.1$ | $89.2\pm0.2$ | $88.4\pm0.4$ | $87.0\pm1.1$ | |
| | | target acc | $81.0\pm0.3$ | $76.0\pm0.2$ | $71.4\pm0.6$ | $64.5\pm0.5$ | $57.0\pm0.5$ | |
| | SAM | concept acc | $92.3\pm0.7$ | $90.7\pm0.4$ | $89.4\pm0.0$ | $88.2\pm0.4$ | $86.5\pm0.6$ | **+0.1** |
| | | target acc | $81.5\pm0.1$ | $77.1\pm0.3$ | $71.0\pm0.7$ | $65.0\pm0.6$ | $57.7\pm0.6$ | **+0.5** |
| Combined (X → C, C → Y) | SGD | concept acc | $91.9\pm0.7$ | $85.9\pm0.5$ | $78.4\pm0.6$ | $67.6\pm1.2$ | $57.3\pm0.3$ | |
| | | target acc | $81.4\pm0.1$ | $75.2\pm0.3$ | $69.2\pm0.5$ | $59.8\pm0.3$ | $50.1\pm0.5$ | |
| | SAM | concept acc | $92.2\pm0.5$ | $86.0\pm0.2$ | $78.5\pm0.1$ | $68.0\pm0.8$ | $57.9\pm0.3$ | **+0.3** |
| | | target acc | $81.4\pm0.6$ | $76.1\pm0.4$ | $69.9\pm0.6$ | $60.8\pm0.4$ | $50.6\pm1.5$ | **+0.6** |

Table 6: Comparison of test accuracy between SGD and SAM on the AWA2 dataset.

| Noise Loc | Optimizer | Metric | Noise Rate | | | | | $\Delta$ |
|---|---|---|---|---|---|---|---|---|
| | | | $nr = 0\%$ | $nr = 10\%$ | $nr = 20\%$ | $nr = 30\%$ | $nr = 40\%$ | |
| *Independent Bottleneck* | | | | | | | | |
| Concept (X → C) | SGD | concept acc | $78.6_{\pm1.1}$ | $78.2_{\pm0.6}$ | $78.1_{\pm0.7}$ | $77.3_{\pm0.5}$ | $75.2_{\pm0.8}$ | |
| | | target acc | $87.1_{\pm2.0}$ | $85.0_{\pm0.2}$ | $83.5_{\pm1.2}$ | $77.9_{\pm1.2}$ | $43.1_{\pm2.8}$ | |
| | SAM | concept acc | $79.0_{\pm0.9}$ | $78.5_{\pm0.6}$ | $78.4_{\pm0.5}$ | $77.8_{\pm0.8}$ | $76.0_{\pm1.1}$ | **+0.5** |
| | | target acc | $87.6_{\pm0.9}$ | $87.2_{\pm1.4}$ | $86.2_{\pm0.2}$ | $81.2_{\pm0.7}$ | $47.7_{\pm3.1}$ | **+2.7** |
| Target (C → Y) | SGD | concept acc | $78.5_{\pm0.9}$ | $78.3_{\pm0.9}$ | $78.3_{\pm1.0}$ | $78.5_{\pm0.9}$ | $78.4_{\pm1.0}$ | |
| | | target acc | $86.4_{\pm2.1}$ | $86.0_{\pm1.8}$ | $85.4_{\pm1.9}$ | $84.9_{\pm2.1}$ | $84.4_{\pm1.9}$ | |
| | SAM | concept acc | $78.8_{\pm0.8}$ | $78.9_{\pm0.9}$ | $78.7_{\pm0.8}$ | $78.7_{\pm0.9}$ | $78.8_{\pm0.8}$ | **+0.5** |
| | | target acc | $87.3_{\pm0.5}$ | $88.1_{\pm0.8}$ | $87.2_{\pm1.5}$ | $85.4_{\pm0.5}$ | $85.0_{\pm1.7}$ | **+1.2** |
| Combined (X → C, C → Y) | SGD | concept acc | $78.5_{\pm0.8}$ | $78.4_{\pm0.8}$ | $78.1_{\pm0.7}$ | $77.3_{\pm0.3}$ | $75.3_{\pm0.8}$ | |
| | | target acc | $86.5_{\pm0.9}$ | $85.5_{\pm0.3}$ | $82.3_{\pm1.4}$ | $77.1_{\pm0.5}$ | $41.9_{\pm1.0}$ | |
| | SAM | concept acc | $78.8_{\pm0.8}$ | $78.6_{\pm0.7}$ | $78.5_{\pm0.5}$ | $77.9_{\pm0.7}$ | $75.9_{\pm1.3}$ | **+0.4** |
| | | target acc | $87.8_{\pm0.8}$ | $88.1_{\pm0.1}$ | $85.7_{\pm0.4}$ | $78.6_{\pm2.8}$ | $46.5_{\pm1.4}$ | **+2.7** |
| *Sequential Bottleneck* | | | | | | | | |
| Concept (X → C) | SGD | concept acc | $78.6_{\pm1.1}$ | $78.2_{\pm0.6}$ | $78.1_{\pm0.7}$ | $77.3_{\pm0.5}$ | $75.2_{\pm0.8}$ | |
| | | target acc | $89.2_{\pm0.6}$ | $87.9_{\pm0.4}$ | $86.1_{\pm0.3}$ | $82.2_{\pm0.9}$ | $73.8_{\pm1.0}$ | |
| | SAM | concept acc | $79.0_{\pm0.9}$ | $78.5_{\pm0.6}$ | $78.4_{\pm0.5}$ | $77.8_{\pm0.8}$ | $76.0_{\pm1.1}$ | **+0.5** |
| | | target acc | $90.8_{\pm0.1}$ | $89.4_{\pm1.1}$ | $88.6_{\pm0.2}$ | $83.2_{\pm4.2}$ | $74.8_{\pm1.2}$ | **+1.5** |
| Target (C → Y) | SGD | concept acc | $78.5_{\pm0.8}$ | $78.3_{\pm0.9}$ | $78.3_{\pm1.0}$ | $78.5_{\pm0.9}$ | $78.4_{\pm1.0}$ | |
| | | target acc | $88.5_{\pm0.6}$ | $87.3_{\pm0.3}$ | $86.6_{\pm0.6}$ | $86.6_{\pm1.2}$ | $83.8_{\pm1.1}$ | |
| | SAM | concept acc | $78.8_{\pm0.8}$ | $78.9_{\pm0.9}$ | $78.7_{\pm0.8}$ | $78.7_{\pm0.9}$ | $78.8_{\pm0.8}$ | **+0.4** |
| | | target acc | $90.5_{\pm0.3}$ | $89.2_{\pm0.9}$ | $90.0_{\pm0.5}$ | $87.7_{\pm0.4}$ | $84.3_{\pm0.8}$ | **+1.8** |
| Combined (X → C, C → Y) | SGD | concept acc | $78.5_{\pm0.8}$ | $78.4_{\pm0.8}$ | $78.1_{\pm0.7}$ | $77.3_{\pm0.3}$ | $75.3_{\pm0.8}$ | |
| | | target acc | $88.7_{\pm0.2}$ | $87.6_{\pm0.3}$ | $85.8_{\pm0.3}$ | $81.8_{\pm1.1}$ | $70.1_{\pm3.9}$ | |
| | SAM | concept acc | $78.8_{\pm0.8}$ | $78.6_{\pm0.7}$ | $78.5_{\pm0.5}$ | $77.9_{\pm0.7}$ | $75.9_{\pm1.3}$ | **+0.4** |
| | | target acc | $90.5_{\pm0.4}$ | $89.5_{\pm0.5}$ | $88.0_{\pm0.5}$ | $82.6_{\pm3.1}$ | $69.6_{\pm6.3}$ | **+1.2** |
| *Join Bottleneck* | | | | | | | | |
| Concept (X → C) | SGD | concept acc | $77.9_{\pm0.5}$ | $77.4_{\pm0.0}$ | $77.4_{\pm0.0}$ | $76.8_{\pm0.1}$ | $73.9_{\pm0.2}$ | |
| | | target acc | $88.9_{\pm0.2}$ | $88.3_{\pm0.1}$ | $89.5_{\pm0.1}$ | $89.4_{\pm0.1}$ | $89.6_{\pm0.1}$ | |
| | SAM | concept acc | $77.7_{\pm0.8}$ | $77.5_{\pm0.7}$ | $77.2_{\pm0.8}$ | $76.7_{\pm0.5}$ | $74.5_{\pm0.6}$ | **+0.0** |
| | | target acc | $91.4_{\pm0.1}$ | $91.9_{\pm0.1}$ | $92.1_{\pm0.1}$ | $92.2_{\pm0.0}$ | $92.3_{\pm0.2}$ | **+2.9** |
| Target (C → Y) | SGD | concept acc | $77.7_{\pm0.6}$ | $77.7_{\pm0.1}$ | $74.2_{\pm0.5}$ | $74.1_{\pm0.5}$ | $73.2_{\pm0.5}$ | |
| | | target acc | $88.9_{\pm0.1}$ | $84.7_{\pm0.4}$ | $82.5_{\pm0.3}$ | $81.2_{\pm0.6}$ | $80.9_{\pm0.4}$ | |
| | SAM | concept acc | $78.0_{\pm0.7}$ | $76.4_{\pm0.4}$ | $75.4_{\pm0.7}$ | $75.1_{\pm0.7}$ | $74.8_{\pm0.6}$ | **+0.0** |
| | | target acc | $91.8_{\pm0.2}$ | $88.8_{\pm0.2}$ | $87.7_{\pm0.2}$ | $86.7_{\pm0.3}$ | $85.8_{\pm0.5}$ | **+4.5** |
| Combined (X → C, C → Y) | SGD | concept acc | $77.8_{\pm0.5}$ | $74.2_{\pm0.4}$ | $70.1_{\pm0.8}$ | $65.4_{\pm0.3}$ | $57.4_{\pm0.2}$ | |
| | | target acc | $88.9_{\pm0.1}$ | $84.2_{\pm0.1}$ | $83.0_{\pm0.3}$ | $82.2_{\pm0.1}$ | $81.7_{\pm0.3}$ | |
| | SAM | concept acc | $78.0_{\pm0.4}$ | $74.9_{\pm0.4}$ | $72.7_{\pm0.6}$ | $67.7_{\pm0.7}$ | $58.9_{\pm0.9}$ | **+0.6** |
| | | target acc | $91.9_{\pm0.3}$ | $89.0_{\pm0.1}$ | $88.4_{\pm0.2}$ | $87.3_{\pm0.2}$ | $86.6_{\pm0.3}$ | **+4.6** |

## D.2 COMPARISON OF TRAINING PROGRESS BETWEEN SGD AND SAM

In Figure 14, 15, and 16, we provide the overall validation accuracy during training for the `Ind`, `Seq`, and `Joi` type models, trained with SGD and SAM on the CUB dataset, across noise rates ranging from 0% to 40%. Overall, while the $g$ model trained with SGD tends to overfit to concept noise, the $g$ model trained with SAM shows better generalization by mitigating the effects of noise and maintains more stable validation accuracy throughout the training process. For $f$ model, since it is a linear model, SAM did not exhibit significant effects. However, it improved the $g$ model, and thus, it ultimately enhanced the training of the $f$ model, as demonstrated by the `Seq` type and `Joi` type.

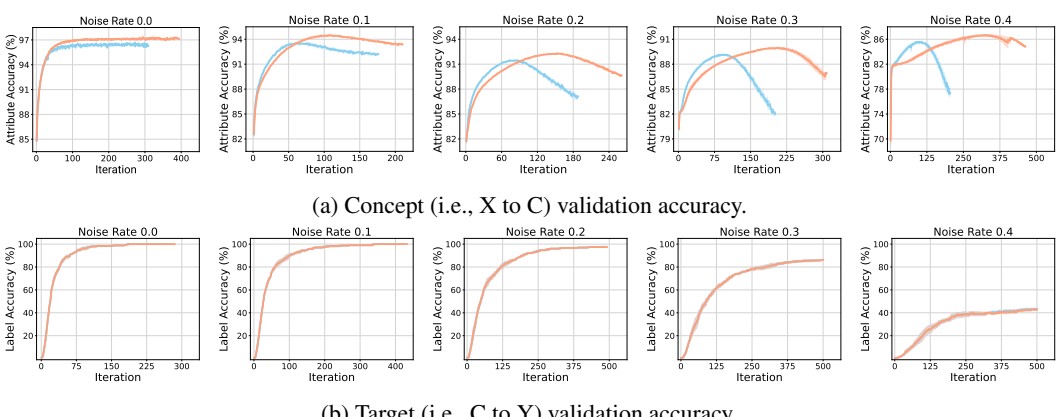

(a) Concept (i.e., X to C) validation accuracy.

(b) Target (i.e., C to Y) validation accuracy.

Figure 14: Results on `Ind`. under concept and target noise with noise rate $00\%$ (left) - $40\%$ (right).

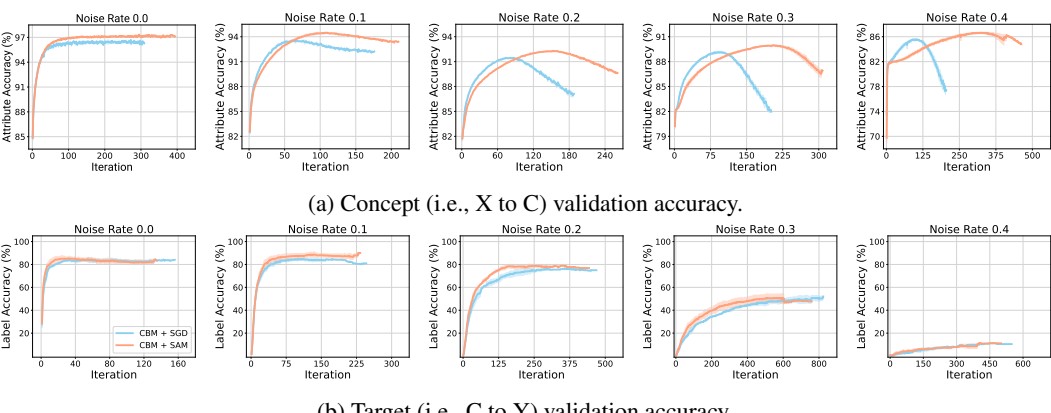

(a) Concept (i.e., X to C) validation accuracy.

(b) Target (i.e., C to Y) validation accuracy.

Figure 15: Results on `Seq`. under concept and target noise with noise rate $00\%$ (left) $- 40\%$ (right).

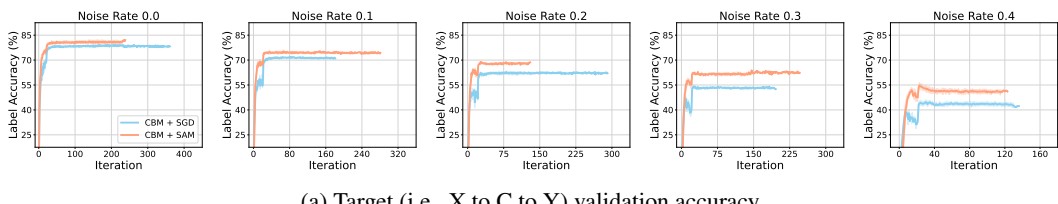

(a) Target (i.e., X to C to Y) validation accuracy.

Figure 16: Results on `Joi`. under concept and target noise with noise rate $00\%$ (left) $- 40\%$ (right).

# E COMPARING INDIVIDUAL CONCEPT ACCURACY: SGD VS. SAM

We analyze the individual concept accuracy of CBMs trained with SGD and SAM, as illustrated in Figure 17. Our findings reveal that SAM consistently enhances individual concept accuracy across both clean and 40% noise conditions. Notably, the improvement is more pronounced under the 40% noise setting, demonstrating the effectiveness of SAM in mitigating label noise and maintaining more reliable concept predictions. This highlights SAM's ability to better preserve the integrity of concept representations even in noisy environments.

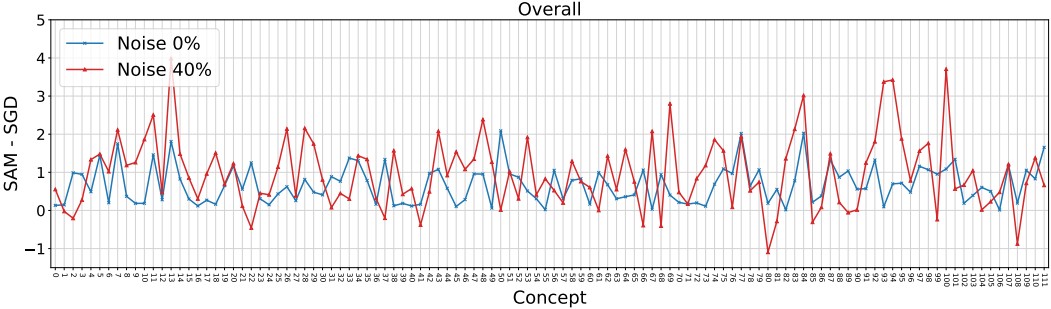

Figure 17: Impact of noise on concept predict accuracy with SGD vs. SAM. The result shows the difference of individual concept prediction accuracy ($SAM_{acc} - SGD_{acc}$) under noise rate of 0% (blue line) and noise rate of 40% (red line).

# F MITIGATING THE LABEL NOISE USING LABEL SMOOTHING

**Label smoothing overview.**     Label smoothing is commonly used to improve the performance of various deep learning models (Szegedy et al., 2016a; Pereyra et al., 2017; Vaswani, 2017; Müller et al., 2019). It computes the loss not with the "hard" targets from the dataset, but with the "soft" target which is a weighted mixture of the targets with a uniform distribution:

$$y_i^r := (1 - r) \cdot y_i + \frac{r}{K} \cdot 1, \tag{1}$$

Here, $y$ refers to the one-hot vector of the hard label, $K$ is the number of classes, and $r$ is the smoothing rate in the range $[0, 1]$. Label smoothing has been shown to prevent overconfidence and improve generalization (Lukasik et al., 2020). It also enhances model robustness to label noise by reducing confidence in noisy labels, similar to the effects of shrinkage regularization.

Table 7: Comparison of test accuracy between base with label smoothing on the CUB dataset.

| Noise Location | Optimizer | Metric | Symmetric Noise | | | Pairwise Noise | | | Δ |
| --- | --- | --- | --- | --- | --- | --- | --- | --- | --- |
| | | | $nr = 0.0$ | $nr = 0.2$ | $nr = 0.4$ | $nr = 0.0$ | $nr = 0.2$ | $nr = 0.4$ | |
| *Independent Bottleneck* | | | | | | | | | |
| Combined (X → C, C → Y) | SGD | concept acc | 96.5±0.1 | 91.6±0.0 | 85.4±0.1 | 96.6±0.1 | 91.5±0.1 | 73.7±0.2 | |
| | | target acc | 74.3±0.3 | 50.4±0.7 | 4.0±0.7 | 74.7±0.6 | 49.4±1.6 | 6.2±1.6 | |
| | Label smoothing | concept acc | 96.6±0.1 | 91.7±0.1 | 85.5±0.0 | 96.6±0.1 | 91.5±0.1 | 73.7±0.3 | +0.0 |
| | | target acc | 74.3±0.2 | 49.6±0.5 | 4.1±0.1 | 75.0±0.2 | 49.1±1.1 | 6.7±1.5 | +0.0 |
| *Sequential Bottleneck* | | | | | | | | | |
| Combined (X → C, C → Y) | SGD | concept acc | 96.5±0.1 | 91.6±0.0 | 85.4±0.1 | 96.6±0.1 | 91.5±0.1 | 73.7±0.2 | |
| | | target acc | 74.2±0.2 | 59.4±0.6 | 6.1±2.6 | 74.8±0.2 | 57.5±0.9 | 7.4±7.0 | |
| | Label smoothing | concept acc | 96.6±0.0 | 91.6±0.0 | 85.4±0.1 | 96.6±0.1 | 91.5±0.1 | 73.7±0.3 | +0.0 |
| | | target acc | 74.7±0.1 | 63.0±0.3 | 16.4±6.8 | 75.0±0.5 | 65.4±1.5 | 14.9±12.0 | +5.0 |
| *Joint Bottleneck* | | | | | | | | | |
| Combined (X → C, C → Y) | SGD | concept acc | 91.9±0.7 | 78.4±0.6 | 57.3±0.3 | 92.2±0.3 | 78.9±0.8 | 56.6±0.4 | |
| | | target acc | 81.4±0.1 | 69.2±0.5 | 50.1±0.5 | 81.3±0.1 | 69.9±0.4 | 49.7±0.4 | |
| | Label smoothing | concept acc | 92.0±0.2 | 79.3±0.3 | 57.8±0.2 | 92.3±0.5 | 79.1±0.7 | 56.5±0.5 | +1.2 |
| | | target acc | 81.7±0.3 | 71.1±0.4 | 53.2±1.5 | 81.5±0.3 | 70.8±0.1 | 50.7±0.5 | +2.1 |

**Results.**     We investigate the effectiveness of label smoothing in CBM under noisy label conditions in both symmetric and pairwise noise. Specifically, we smooth both concept labels with $r = 0.001$ and class labels $r = 0.1$ during training. We find that label smoothing does not show significant

improvement for `Ind` type, where the averaged improvement was 0%. However, for `Seq` and `Joi`, we find that as the noise increases, the improvement in performance tends to become even more pronounced. Specifically, we find that when noise occurs in the `Seq` model, label smoothing effectively enhances performance the most. At a noise rate of 40%, the target accuracy in both symmetric and pairwise noise settings is more than double compared to the existing baseline. Overall, these results indicate that label smoothing effectively mitigates the detrimental effects of label noise.

# G EXPERIMENTAL DETAILS

## G.1 DATASET

Overall, Each example is a triplet of (image $x$, concepts $c$, target $y$) corresponding to a target class, where the concepts have binary values: 1 for true or 0 for false.

**CUB** CUB (Wah et al., 2011) is a standard dataset commonly used to study Concept Bottleneck Models (Koh et al., 2020; Zarlenga et al., 2022; Xu et al., 2024), consisting of 5,994 training examples and 5,794 test examples, with input images of $224 \times 224$ pixels. Following the original work (Koh et al., 2020), the final dataset includes only 112 of the 312 most prevalent binary attributes.

**AWA2** AWA2 (Xian et al., 2018) is a zero-shot learning dataset consisting of 37,322 images across 50 classes, with input images of $224 \times 224$ pixels. Each sample is associated with 85 binary concepts.

## G.2 TRAINING

For our CBM training process, we utillize InceptionV3 (Szegedy et al., 2016b) as the backbone, pre-trained on ImageNet (Deng et al., 2009) and subsequently fine-tuned on the CUB (Wah et al., 2011) and AWA2 (Xian et al., 2018) dataset. In line with previous work Koh et al. (2020); Xu et al. (2024), we select the 112 concepts for training CUB dataset, and 85 concepts for training AWA2 dataset. We follow to the preprocessing techniques outlined by Koh et al. (2020), applying data augmentation to each training image with random color jittering, horizontal flipping, and cropping to a resolution of 224. During inference, images are center-cropped and resized to 224 pixels.

The `Ind` and `Seq` models are trained using a learning rate of 0.01, while the `Joi` model was set to 0.001. We set learning rate schedules, reducing it by a factor of 10 every 10, 15, or 20 epochs until it reaches 0.0001. A regularization strength of 0.0004 is used, and model selection is based on the highest validation accuracy.

Training is conducted using a batch size of 64, with the optimizer being SGD with a momentum of 0.9 for all models except those trained with SAM, where we use a sharpness parameter $\rho$ set to 0.1 for `Ind` and `Seq` and `Joi` for 0.01 by grid search over $[0.01, 0.05, 0.1]$. For bottleneck models, each concept's contribution to the overall loss is weighted equally. For the `Joi` moddel, the task-concept trade-off hyperparameter is guided by $\lambda$ set as 0.001 by grid search over $[0.1, 0.01, 0.005, 0.001]$ in the overall noise setting. Additionally, binary cross-entropy loss for each concept prediction is adjusted for class imbalance, following the normalization approach in Koh et al. (2020) This approach ensure that the models were rigorously trained, balancing between target accuracy and concept interpretability across various training strategies.

