# OpenReview forum: "Concept Bottleneck Models under Label Noise"
_ICLR.cc/2025/Conference — Submitted to ICLR 2025_

### Official Review · Reviewer_6yJN · 2024-10-30

**Soundness:** 2
**Presentation:** 3
**Contribution:** 2
**Rating:** 3
**Confidence:** 4

**Summary:**

The article investigates concept bottleneck models as regards their brittleness to label noise in the involved concepts. The article shows that this is indeed the case, using different amount of noise and benchmark data sets. It investigates the impact on both, meaningfulness of concepts and accuracy, as well as visualizes this. Then it shows that a technology proposed for mitigating sensitivity to label noise helps (somewhat).

**Strengths:**

The article investigates a novel and relevant problem which, to my knowledge, has not been targeted yet. The article is extensive w.r.t. evaluation. It proposes a first mitigation.

**Weaknesses:**

The article is merely experimental and does not provide theoretical insights. It widely ignores previous work on dealing with label noise such as probabilistic treatments, differences of sensitivity based on the cost function etc. (see eg overviews by Ata Kaban and Benoit Frenay). The expertiments are somewhat lengthy with not so much insight. The mediation procedure is only mildly successful.

**Questions:**

What is the effect of asymmetries of label noise?
Can you investigate different cost functions for training and different other methods?
How does this relate to technologies which do not assume availability of information w.r.t. concepts for all data?
How does the effect relate to aspects of the suitability of concepts?

---

> ### Comment · Reviewer_6yJN · 2024-11-25
> **acknowledgement of replies**
>
> I have read all reviews and replies to it. It seems that there is some agreement of the reviewers, actually, which did not change much within the discussion

---

> > ### Author Response · Authors · 2024-11-26
> > **Response to 6yJN**
> >
> > We sincerely appreciate your interest in our work and for providing important feedback. We would be very grateful if you could review our responses we have prepared.
> >
> > **Theoretical insights?**
> > > The article is merely experimental and does not provide theoretical insights.
> >
> > With all due respect, we disagree with the reviewer’s criticism on not presenting theoretical results (and by implication, trivial). We firmly believe that empirical studies can also render valuable insights into understanding the complex nature of deep learning. In our work, we conducted comprehensive systematic experiments to uncover the previously unknown issue with CBMs, and presented potential alternatives that could mitigate the issue effectively right off in practice. There are myriads of works in the literature that have been well recognized by the community without presenting a single theory. In this regard, we strongly request the reviewer to reconsider this seemingly unfair criticism.
> >
> > ---
> >
> > **Previous work to deal with label noise?**
> > > It widely ignores previous work on dealing with label noise such as probabilistic treatments, differences of sensitivity based on the cost function etc. (see eg overviews by Ata Kaban and Benoit Frenay).
> >
> > It is well-known that when training a deep model under noisy conditions, the model initially learns from clean and noise data effectively. However, as training progresses, it begins to fit the noisy data, which often has relatively higher loss values, leading to overfitting on the noisy data (see [1]). In this context, recent study has shown that SAM enables the model to learn clean data stably while preventing the loss from noisy data from growing excessively. For more detail, please refer to our response for Reviewer 6SJD titled "Theoretical intuition on SAM".
> >
> > As highlighted in your suggested paper [2], there are various approaches to addressing label noise, such as label noise-robust models, data cleansing methods, and label noise-tolerant learning algorithms. Among these, we focused on label noise-tolerant learning algorithms and chose SAM as it has been shown in recent study to effectively address the aforementioned issues in deep models. Through our experiments, we confirmed that SAM effectively mitigates the impact of label noise.
> >
> > Additionally, we want to highlight that in the Appendix, we have also included results for label smoothing, a well-known method for mitigating label noise [3], as an additional approach.
> >
> > ---
> >
> > **Exploration of different mitigation methods**
> > > Can you investigate different cost functions for training and different other methods?
> >
> > Thank you for your suggestion. We have included experimental results using label smoothing, a widely used method for mitigating label noise, in the Appendix. As there are various approaches to mitigating label noise, we would appreciate it if you could kindly suggest any specific methods you would like us to explore beyond those we have presented. We will conduct the requested experiments and provide a report.
> >
> > ---
> >
> > **Q. Effect of asymmetries of label noise?**
> > > What is the effect of asymmetries of label noise?
> >
> > Thank you for raising this point. The pairwise noise (i.e., asymmetric noise) provides a realistic representation of label noise by incorporating dependencies between labels. For instance, it accounts for the scenarios where confusing a dog with a cat is more likely than confusing a dog with a flower. This setting can capture an additional aspect of real-world noise scenarios [4].
> >
> > ---
> >
> > **Q. Cases where not all concept information is available?**
> > > How does this relate to technologies which do not assume availability of information w.r.t. concepts for all data?
> >
> > Thank you for your question. To the best of our knowledge, there are studies that deal with scenarios where concepts are not available for all data, such as semi-supervised CBM [5]. While the suggested setting could be an interesting direction for future work, our study, as stated in Section 2, focuses on a supervised learning setting. Therefore, this question lies beyond the scope of our current work.
> >
> > ---
> >
> > **Q. Suitability of concepts?**
> > > How does the effect relate to aspects of the suitability of concepts?
> >
> > Thank you for your question. As we are not sure if we have fully understood the question, we would appreciate it if you could provide further clarification.
> >
> > ---
> >
> > [1] Zhang, Chiyuan, et al. "Understanding deep learning (still) requires rethinking generalization." ACM, 2021.
> > [2] Frénay, Benoît, and Ata Kabán. "A comprehensive introduction to label noise." ESANN. 2014.\
> > [3] Lukasik, Michal, et al. "Does label smoothing mitigate label noise?." PMLR, 2020.\
> > [3] Scott, Clayton, et al. “Classification with asymmetric label noise: Consistency and maximal denoising.” JMLR, 2013.\
> > [4] Hu, Lijie, et al. "Semi-supervised Concept Bottleneck Models." arXiv preprint arXiv:2406.18992, 2024.

---

> > > ### Comment · Reviewer_6yJN · 2024-11-26
> > > **appreciation of reply**
> > >
> > > I appreciate the reply of the authors. Suitability of concepts: I have been asking about whether these can account for concepts which correspond to a semantic meaning (which could be tested e.g. using explicit synthetic data generated by contexts).
> > >
> > > I appreciate that some results deal with asymmetries and label smoothing as classical concept.
> > >
> > > I appreciate that there are valid contributions which do not provide theory, surprising, impressive  or extremely useful results. I do not agree that a systematic study without theoretical insight or novel concepts is sufficient for A* conferences.

---

> > > > ### Author Response · Authors · 2024-12-02
> > > > **Thank you**
> > > >
> > > > Thank you for your response. Regarding your question about the suitability of concepts for semantic meanings, we believe this falls outside the scope of the experiments and objectives we set for this work. While we recognize that this is a valuable discussion that broadens the perspective of a study, it is beyond the scope of our work.
> > > >
> > > > We respectfully disagree with the comment that our work “does not provide surprising, impressive or extremely useful results.” While we acknowledge that our theoretical contributions may be somewhat limited, our extensive empirical analyses demonstrate some specific vulnerability and robustness of CBMs, which was largely overlooked in prior studies. This is quite unexpected since models under label noise may intuitively be expected to degrade model performance. We believe our work highlights a key challenge in training CBMs under label noise, and addressing this issue is crucial for making CBMs more practical and effective in real-world applications.
> > > >
> > > > We understand that assessments can vary depending on perspective, but we feel this particular criticism does not fully reflect the broader strengths of our work. Nevertheless, we greatly appreciate your thoughtful feedback.

---

> > > > > ### Comment · Reviewer_6yJN · 2024-12-02
> > > > > **Reply and Appreciation**
> > > > >
> > > > > Thanks for the comment, I appreciate the effort. It seems we simply disagree here.

---

> > > > > > ### Author Response · Authors · 2024-12-04
> > > > > > **Appreciation for the reply**
> > > > > >
> > > > > > We sincerely thank you for your thoughtful comment and the effort you have dedicated.

---

### Official Review · Reviewer_kFmQ · 2024-11-02

**Soundness:** 2
**Presentation:** 2
**Contribution:** 1
**Rating:** 3
**Confidence:** 5

**Summary:**

The paper has a clear motivation to investigate and demonstrate the severity of detrimental effects caused by label noise of different levels, specifically examining how label noise compromises the performance of CBM. Specifically, the paper utilizes SAM to mitigate these impacts. Empirical results show that SAM has the potential to significantly enhance the robustness of CBM, compared to SGD.

**Strengths:**

- This paper has a clear motivation focused on the robustness under conditions with label noise.
- From an overall view, this paper is easy to follow and the idea of using SAM is reasonable.
- It provides detailed insights into the underlying causes of decreased prediction accuracy, both for intermediate concepts and final targets, contributing to a deeper understanding of the effects of label noise.

**Weaknesses:**

- Although the paper details the impact and offers an in-depth analysis of label noise, the methodology could be strengthened. The use of SAM, which is borrowed from prior works, is the sole approach for enhancing robustness.
- The emphasis on label noise impact might be overstated, especially as noise levels are extended to 40%, which is unconventional for real-world scenarios. Natural label noise is already present in datasets like CUB and AWA2.
- The proposed method's improvement over joint CBM is relatively incremental, even under high noise levels, which may limit the perceived effectiveness of SAM.
- In Table 1, the concept prediction accuracy is only 92.4% for joint CBM when the noise level is 0%, which does not correspond to 96.9% reported in the vanilla CBM literature, despite using the same Inception-V3 backbone. What might account for this discrepancy?
- This paper lacks an alternative type of literature such as [1] that generates concepts using LLM/VLM. This category is highly relevant to the label noise setting.
- The vanilla CBM is the only baseline and more baselines should be considered to validate the effectiveness.
- The authors discuss the impacts on interpretability by evaluating concept prediction accuracy. However, these two definitions are not strictly synonymous.

[1] Oikarinen, Tuomas, et al. "Label-free Concept Bottleneck Models." International Conference on Learning Representations. 2023.

**Questions:**

- In Fig. 7, I noticed a sharp drop in target validation accuracy for the SGD training methods after a certain number of iterations, yet the target validation accuracy appears to stabilize overall. Could you provide an explanation for this observation?
- Also, for figure-7-b, why are the performance of SGD and SAM almost the same?

---

> ### Author Response · Authors · 2024-11-25
> **Response to kFmQ (1/2)**
>
> We appreciate the reviewer for acknowledging the clear motivation of our method, and giving us constructive feedback. While we respond to the reviewer’s specific comments as below, we would be keen to engage in any further discussion.
>
> ---
>
> **Beyond SAM?**
> > Although the paper details the impact and offers an in-depth analysis of label noise, the methodology could be strengthened. The use of SAM, which is borrowed from prior works, is the sole approach for enhancing robustness.
>
> Thank you for the constructive criticism. Our primary focus was to highlight the vulnerability of CBM to label noise and to propose an effective remedy to mitigate this problem. The reason for adopting SAM (and additionally, label smoothing, as discussed in the appendix) is not arbitrary; rather, it is based on the fact that these methods are among the most effective approaches for addressing label noise as supported by prior works [1, 2]. Through various experiments, we confirm their effectiveness, e.g., we observe that even slight improvements in concept prediction caused by SAM can lead to significant gains in target prediction performance. We recognize that a broader comparison with alternative methods would strengthen the study and plan to pursue this in future work.
>
> ---
>
> **Unrealistic noise setting**
> > The emphasis on label noise impact might be overstated, especially as noise levels are extended to 40%, which is unconventional for real-world scenarios. Natural label noise is already present in datasets like CUB and AWA2.
>
> Thank you for raising this concern. While we agree that a 40% noise level is quite high, our analysis is not confined to this level, and in fact, it covers a wide range of noise levels ranging from 0% to 40%. Therefore, we would respectfully disagree with the assertion that the impact of label noise is overstated.
>
> Notably, we observe that even with a modest noise level of 10%—a more realistic scenario in practice—the target accuracy of CBMs drops significantly from 74.3% to 61.9%. This behavior is distinctly different from traditional end-to-end models, highlighting the unique vulnerability of CBM to concept noise. Therefore, our claims about the impact of noise are reasonable and well-supported..
>
> ---
>
> **Accuracy**
> > In Table 1, the concept prediction accuracy is only 92.4% for joint CBM when the noise level is 0%, which does not correspond to 96.9% reported in the vanilla CBM literature, despite using the same Inception-V3 backbone. What might account for this discrepancy?
>
> Thank you for pointing this out. In the joint model, the model minimizes the following objective:
>
> $\hat{f}, \hat{g} = \arg\min_{f,g} \sum_i \left[ L_Y \left( f \left( g \left( x^{(i)} \right) \right), y^{(i)} \right) \right] + \sum_j \lambda \mathcal{L}_{C_j} \left( g \left( x^{(i)} \right); c^{(i)} \right) \quad \text{for some } \lambda > 0.$
>
> Instead of directly adopting the $\lambda$ value from prior work, we tune hyperparameters for each clean and noise settings, with a primary focus on target accuracy. As a result of using different $\lambda$ value, the concept accuracy differs from that reported in the original work.
>
> ---
>
> **LLM/VLM-based concept**
> > This paper lacks an alternative type of literature such as [2] that generates concepts using LLM/VLM. This category is highly relevant to the label noise setting.
>
> Thank you for your suggestion. Previous studies, including [2], have shown that even when concepts are annotated using LLMs, incorrect or ambiguous labels remain unavoidable. This implies that noisy labels are quite inevitable even when using LLM-based annotation approaches.
>
> Investigating different types of noise including ones created by LLMs is certainly an interesting direction to pursue. We will discuss this aspect in the revised version.

---

> > ### Author Response · Authors · 2024-11-25
> > **Response to kFmQ (2/2)**
> >
> > **Correlation of Interpretability and Concept prediction accuracy**
> > > The authors discuss the impacts on interpretability by evaluating concept prediction accuracy. However, these two definitions are not strictly synonymous.
> >
> > Thank you for the comment. We agree that the concept prediction accuracy and interpretability are not strictly equivalent. However, we believe there is a meaningful connection between the two. Concept prediction accuracy reflects how well the concept representations learned by the model align with the ground-truth labels. Higher concept accuracy suggests that the model has effectively captured and represented the underlying concepts, which contributes to interpretability.
> >
> > Additionally, prior studies, such as those using Concept Alignment Score [4], have also linked alignment of learned concept representations with ground-truth concepts to interpretability. Since interpretability is inherently defined as the degree to which one can understand, we believe there is no single universally accepted definition or evaluation method. Therefore, we see concept prediction accuracy as one of many reasonable proxies for assessing it. We appreciate this constructive feedback and will take it into consideration to further refine the discussion.
> >
> > ---
> >
> > **Q. Training process of SAM and SGD**
> > > In Fig. 7, I noticed a sharp drop in target validation accuracy for the SGD training methods after a certain number of iterations, yet the target validation accuracy appears to stabilize overall. Could you provide an explanation for this observation? \
> > > Also, for figure-7-b, why are the performance of SGD and SAM almost the same?
> >
> > Thank you for the detailed questions. The difference in stabilization between concept prediction accuracy and target prediction accuracy can be attributed to the structural differences between the two models. The concept prediction model, being a deep network, initially learns both clean data and noise effectively (See [4]). However, as training progresses, it overfits to the noise, resulting in a decline in validation accuracy (i.e., performance on the clean dataset).
> >
> > In contrast, the target prediction model, being a linear (or shallow) network, inherently lacks the capacity to fit noise. As a result, its performance remains stable on the clean dataset, giving the appearance of stable target accuracy overall.
> >
> > While precisely explaining these distinct training and generalization dynamics is beyond the scope of this work, our experiments clearly demonstrate that SAM helps to achieve significantly more robust concept predictions compared to SGD, particularly under noisy conditions, by mitigating the effects of noise during training.
> >
> > Moreover, the marginal difference in target prediction accuracy between SAM and SGD can be attributed to the target prediction, i.e., a linear model, is not overparameterized. Recent study [6] demonstrates that benefits of SAM are most pronounced in overparameterized models. As a result, both SAM and SGD exhibit similar trends in validation progress for the $f$ model.
> >
> > ---
> >
> > [1] Baek, Christina, et al. "Why is SAM robust to label noise?" ICLR, 2024.\
> > [2] Lukasik, Michal, et al. "Does label smoothing mitigate label noise?." PMLR, 2020.\
> > [3] Oikarinen, Tuomas, et al. "Label-free Concept Bottleneck Models." ICLR, 2023.\
> > [4] Zarlenga, Mateo Espinosa, et al. "Concept embedding models." NeurIPS, 2022.\
> > [5] Zhang, Chiyuan, et al. "Understanding deep learning (still) requires rethinking generalization." ACM, 2021.
> > [6] Shin, Sungbin, et al. "The Effects of Overparameterization on Sharpness-aware Minimization: An Empirical and Theoretical Analysis." arXiv preprint arXiv:2311.17539, 2023.\

---

> > > ### Author Response · Authors · 2024-11-29
> > > **Dear Reviewer**
> > >
> > > Dear Reviewer kFmQ
> > >
> > > We sincerely appreciate your time and thoughtful feedback on our submission. To address your concerns, we have worked diligently to provide detailed clarifications and discussions in our rebuttal. As some time has passed since we submitted our responses, we kindly request that you review our rebuttal to evaluate whether our responses sufficiently address the issues raised, and re-evaluate the score accordingly.
> > >
> > > We are more than happy to address any additional questions or comments you might have.
> > >
> > > Best wishes,
> > > The authors.

---

> ### Comment · Reviewer_kFmQ · 2024-11-29
> **Follow-up to the author rebuttal**
>
> Thank you to the authors for their time and effort in addressing my initial concerns. However, after reviewing their responses, I must express that my concerns have not been fully addressed.
>
> First, while I understand and acknowledge that the choice of SAM is not arbitrary, the methodology remains weak, and the theoretical contributions are limited, which undermines the overall impact of the work. Furthermore, the authors claim that “we observe that even slight improvements in concept prediction caused by SAM can lead to significant gains in target prediction performance,” but this phenomenon seems to only be evident in independent and sequential CBMs. Moreover, there is a lack of ablation studies to rule out the possibility that the observed improvements could be due to other factors, such as the learning rate.
>
> Second, the target accuracy does not align with that of the vanilla CBM across all three models. The authors state that “we tune hyperparameters for each clean and noise setting, with a primary focus on target accuracy,” which could introduce bias and result in unfair comparisons. A particularly perplexing finding is that the target accuracy is lower than that of the vanilla CBM, even though the authors explicitly state that they prioritize target accuracy in their tuning process. Could the authors provide a clearer explanation for this discrepancy?
>
> Third, the authors claim that “the target accuracy of CBMs drops significantly from 74.3% to 61.9%” with a modest noise level of 10%. However, this substantial drop is also observed in the SAM condition, although I do not see results reported for the 10% noise level specifically. Given that a similar issue arises for both SGD and SAM, I question whether SAM is truly beneficial under high noise levels. Could the authors provide more evidence or justification for the effectiveness of SAM in such settings?
>
> Fourth, the authors mentioned that “We will discuss this aspect in the revised version,” but I have not been able to find the corresponding discussion. Could the authors point out where this is addressed? Specifically, aside from the discussion of LLM-based CBMs, the authors also indicated they would explore the correlation between interpretability and concept prediction accuracy, yet this discussion is not present in the revised version.
>
> Best,
> Reviewer kFmQ

---

> > ### Author Response · Authors · 2024-12-04
> > **Follow-up response to kFmQ**
> >
> > Thank you for your detailed feedback. We would like to address your concerns and provide the following clarifications.
> >
> > In our experiments, the target accuracy achieved by our method was 81.4%, compared to 80.1% for the vanilla CBM. This demonstrates that our tuning process does not introduce contradictory results. Additionally, all results for both SGD and SAM across varying conditions are included in Appendix D, as referenced in the main text.
> >
> > Our analysis further reveals that while SAM experiences a similar performance drop under noisy conditions, it consistently demonstrates greater robustness compared to SGD, particularly at the highest noise levels. These findings highlight that SAM achieves improved stability and reliability, even in challenging scenarios.
> >
> > Lastly, we plan to provide a more detailed discussion on LLM-based CBMs in future revisions. We hope these clarifications adequately address your concerns, and we sincerely thank you once again for your thoughtful and comprehensive feedback.

---

### Official Review · Reviewer_6SJD · 2024-11-03

**Soundness:** 2
**Presentation:** 3
**Contribution:** 3
**Rating:** 6
**Confidence:** 4

**Summary:**

The paper is concerned with inherently interpretable models called Concept Bottleneck Models (CBMs). These models require extensive concept labeling; however, these labels are usually assumed to be perfect. The paper explores how noise in these concept labels affects the final target prediction. The authors perform extensive experiments across all three variants of CBMs to show the detrimental effects of concept labels. The authors then proposed SAM training to improve concept and target accuracy.

**Strengths:**

- Label noise and mislabeling are well-motivated problems, especially from a CBM perspective. No work has explicitly examined this problem.

- The paper structure is very intuitive and easy to read.

- The authors showed the degradation due to label noise well with comprehensive experiments.

**Weaknesses:**

- While the experiments are comprehensive for Section 3 and 4. Some of the results are pushed to the appendix (which is fine), however it would have been nice to summarise them in brief in the text.

- I enjoyed reading up to Section 4. Thank you. However, I would have appreciated some theoretical intuition on why SAM works better (unless I missed it).

- The paper performs experiments with CUB and AwA2 datasets, popular benchmark datasets for CBMs. These datasets however have a strong correlation between concept labels and the target label. I can imagine label noise to be very detrimental (as observed from Figure 2).  The potential observed effect due to label noise might be weaker in the case of diverse concepts for each target.

- The paper claims to be first the paper to looking at label noise in CBMs, while I would not refute this, I would like to point out the authors to some very relevant papers - [1] (noise added to concept labels, similar to some of the exps in Sec3) [2]-(concept robustness and adv attacks)

[1] - Sheth, Ivaxi, and Samira Ebrahimi Kahou. "Auxiliary losses for learning generalizable concept-based models." Advances in Neural Information Processing Systems 36 (2024).

[2] - Sinha, Sanchit, et al. "Understanding and enhancing robustness of concept-based models." Proceedings of the AAAI Conference on Artificial Intelligence. Vol. 37. No. 12. 2023.

**Questions:**

- In Sec 6.1 authors show that other CBM variants are also susceptible to label noise. This alings with CBMs, however the obvious question for me is, does SAM training improve the robustness? Why or Why not?

- Concept labeling CBMs is very difficult, there is an increasing interest in using LLMs for concept annotation. Can the authors kindly comment (maybe in Limitations section of the paper), on whether label noise will impact such concept labels.

- Interventions are useful aspect of CBMs. What is the impact of interventions to reduce label noise? I assume interventions may be less effective. Does SAM improve it?


Minor:

- Line 101, now caps for "We".

- Figure 3, which model is used not specified? Joint/Ind/Seq?

---

> ### Author Response · Authors · 2024-11-23
> **Response to 6SJD (1/2)**
>
> We sincerely thank the reviewer for finding our work well-motivated and intuitive. We also appreciate the reviewer’s constructive feedback. While we make clarifications to specific comments below, please let us know if there is anything we need to address further. We would be keen to engage in any further discussion.
>
> ---
>
> **Summerization**
> > While the experiments are comprehensive for Section 3 and 4. Some of the results are pushed to the appendix (which is fine), however it would have been nice to summarise them in brief in the text.
>
> Thank you for your suggestion. We will revise the content to provide a concise summary in the final version.
>
> ---
>
> **Theoretical intuition on using SAM**
> > I enjoyed reading up to Section 4. Thank you. However, I would have appreciated some theoretical intuition on why SAM works better (unless I missed it).
>
> Thank you for your feedback. According to a previous study [1], the robustness of SAM to label noise can be explained theoretically. Specifically, SAM acts like an $L2$-norm penalty on intermediate activations and the weights in the last layer in a binary classification task using a 2-layer network.
>
> In detail, assuming the function $f(x)$ is bounded by $\|f(x)\|_2 \leq C$, the loss $\ell(x)$ for a data point $x$ is bounded as $\log\big(1 + (K - 1) \exp(-C)\big) \leq \ell(x) \leq \log\big(1 + (K - 1) \exp(C)\big).$
>
> As $C$ decreases, the loss for clean data remains at a non-negligible level, while the loss for noisy data is capped at $\log\big(1 + (K - 1) \exp(C)\big)$. This ensures that the model learns clean data stably while preventing the loss from noisy data from growing excessively.
>
> Although this analysis is based on a simple 2-layer linear network, SAM is expected to exert even stronger regularization in deep neural networks due to their larger and more complex weight spaces, thereby enhancing robustness to label noise.
>
> ---
>
> **Correlation of concepts and target**
> > The paper performs experiments with CUB and AwA2 datasets, popular benchmark datasets for CBMs. These datasets however have a strong correlation between concept labels and the target label. I can imagine label noise to be very detrimental (as observed from Figure 2). The potential observed effect due to label noise might be weaker in the case of diverse concepts for each target.
>
> Thank you for suggesting these relevant papers. We will make sure to include these papers in our work.
>
> ---
>
> **Related Works**
> > The paper claims to be first the paper to looking at label noise in CBMs, while I would not refute this, I would like to point out the authors to some very relevant papers - [2] (noise added to concept labels, similar to some of the exps in Sec3) [3]-(concept robustness and adv attacks)
>
> We appreciate the reviewer for suggesting related works [2, 3]. We will include the suggested works in our final version.

---

> > ### Author Response · Authors · 2024-11-23
> > **Response to 6SJD (2/2)**
> >
> > **Q. Impact of LLM based concept labeling**
> > > Concept labeling CBMs is very difficult, there is an increasing interest in using LLMs for concept annotation. Can the authors kindly comment (maybe in Limitations section of the paper), on whether label noise will impact such concept labels.
> >
> > Thank you for the reviewer's thought-provoking question. Previous studies on concept annotation using LLMs have highlighted that even when leveraging LLMs for concept annotation, the incorrect or ambiguous labels are unavoidable [4]. This implies that noisy labels are an inherent issue even when using LLMs for annotation.
> >
> > To address this challenge, it would be valuable to investigate the label noise introduced by LLMs and thoroughly analyze its impact on model performance. We plan to discuss this limitation in the Limitation section and consider it as part of our future work.
> >
> > ---
> >
> > **Q. Effectiveness of intervention?**
> > > Interventions are useful aspect of CBMs. What is the impact of interventions to reduce label noise? I assume interventions may be less effective. Does SAM improve it?
> >
> > We thank the reviewer for the insightful question. To evaluate the impact of interventions in noisy settings, we conducted experiments under the concept noise setting, and the results are provided below. Here, we denote the intervention counts as $ic$.
> >
> > | Optimizer | Noise Ratio | $ic = 0$ | $ic = 5$ | $ic = 10$ | $ic = 15$ | $ic = 20$ | $ic = 25$ |
> > |:--------------:|:-----------:|:-------:|:-------:|:-------:|:-------:|:-------:|:-------:|
> > |SGD | Clean | 74.7 | 79 | 85.3 | 91.7 | 95.5 | 96.9 |
> > |SGD | $nr = 0.1$ | 61.9 | 74.8 | 85.8 | 92.1 | 95.0 | 96.2 |
> > |SAM | $nr = 0.1$ | 65.7 | 78.0 | 87.4 | 92.8 | 95.2 | 96.2 |
> > |SGD | $nr = 0.2$ | 52.0 | 72.3 | 85.4 | 91.1 | 93.9 | 95.4 |
> > |SAM | $nr = 0.2$ | 56.9 | 74.7 | 86.3 | 91.6 | 94.3 | 95.5 |
> > |SGD | $nr = 0.3$ | 33.1 | 56.0 | 72.9 | 82.1 | 87.5 | 90.8 |
> > |SAM | $nr = 0.3$ | 38.1 | 59.0 | 73.9 | 82.2 | 87.4 | 90.8 |
> > |SGD | $nr = 0.4$ | 10.7 | 22.0 | 36.5 | 50.0 | 61.0 | 69.7 |
> > |SAM | $nr = 0.4$ | 11.0 | 22.6 | 37.7 | 50.4 | 60.8 | 69.4 |
> >
> > When focusing on SGD results, we observe that as the noise ratio increases, the impact of interventions becomes more pronounced, showing that a sufficient number (e.g., 20-25) of interventions can bring performance close to that of clean data under moderate noise levels. However, in case of extreme noise settings, this trend does not hold. We hypothesize that this arises mainly from the tendency of models to learn incorrect concept-target relationships under severe noise, which prevents it from effectively leveraging the interventions.
> >
> > Another key finding is that we observe that the performance improvement from increased intervention counts is greater in datasets with higher noise levels, except for cases of extremely high noise. This highlights that even correcting a few incorrect concepts can yield significant performance gains especially in noisy settings.
> >
> > When using SAM, it consistently outperforms the baseline. Since the models trained with SAM can predict more accurate concepts, resulting in higher overall performance compared to those trained with SGD. However, the performance improvement achieved through interventions is greater with SGD than with SAM. We hypothesize that this occurs because as the number of correctly predicted concepts increases, the relative impact of interventions diminishes. Since the models trained with SAM predict more accurate concepts compared to those trained with SGD, the performance gains from interventions are more substantial with SGD.
> >
> > ---
> >
> > [1] Baek, Christina, et al. "Why is SAM robust to label noise?" ICLR, 2024. \
> > [2] Sheth, Ivaxi, et al. "Auxiliary losses for learning generalizable concept-based models." NeurIPS, 2024. \
> > [3] Sinha, Sanchit, et al. "Understanding and enhancing robustness of concept-based models." AAAI, 2023. \
> > [4] Oikarinen, Tuomas, et al. "Label-free Concept Bottleneck Models." ICLR, 2023.

---

> > > ### Comment · Reviewer_6SJD · 2024-11-26
> > > **Thank you**
> > >
> > > Dear authors,
> > >
> > > I want to thank you for replying to my questions/concerns. I especially appreciate intervention results and some theoretical intuition on SAM, I hope they will be part of the revised version. Additionally similar to some of the other reviewers, I would appreciate more discussion on label noise for LLM concept annotations.
> > >
> > > I am happy to increase the score to 6. The reason for not higher - LLM concept seems like a natural exp and weak theoretical motivation, however I think it is an imp problem to consider.

---

> > > > ### Author Response · Authors · 2024-11-26
> > > > **Thank you**
> > > >
> > > > We sincerely thank the reviewer for their thoughtful consideration of our rebuttal and for increasing the score. Your questions and the suggestion to conduct an intervention experiment have greatly contributed to enhancing our work.  We will make sure to reflect these discussions in the final version.
> > > >
> > > > During the remaining rebuttal period, we will try to further explore LLM-based concepts. If you have any additional comments or discussions, please feel free to share them with us.
> > > >
> > > > Lastly, we would like to kindly request your confirmation regarding the updated score, as it has not yet been reflected. Once again, we deeply appreciate your time and effort.

---

> > > > > ### Comment · Reviewer_6SJD · 2024-12-02
> > > > >
> > > > > Apologies, updated the score now.

---

> > > > > > ### Author Response · Authors · 2024-12-04
> > > > > > **Appreciation of the update**
> > > > > >
> > > > > > We sincerely appreciate you taking the time to confirm and for updating the score once again.
> > > > > > Your thoughtful and insightful comments have been instrumental in significantly improving our work.
> > > > > >
> > > > > > Once again, we are truly grateful for the time and thoughtful consideration you have dedicated.

---

### Official Review · Reviewer_To3J · 2024-11-03

**Soundness:** 3
**Presentation:** 3
**Contribution:** 2
**Rating:** 3
**Confidence:** 4

**Summary:**

The submission empirically investigates the effect of injecting label noise in concept bottleneck models for image classification. These models require two types of labels: class labels and concept labels. The effect is evaluated in three training regimes: training the concept predictor and the classifier separately, training them in sequence, and training them jointly. The results on two image classification datasets show that increasing label noise decreases classification accuracy. The effect is smallest for joint training. However, concept prediction accuracy is severely affected in joint training. The submission also shows that sharpness-aware minimization can be used to ameliorate accuracy. Auxiliary results are given for two variants of concept bottleneck models, structured noise, and different network architectures. These support the findings regarding the effects on classification accuracy.

**Strengths:**

The submission provides an extensive set of results investigating the effect of label noise on concept bottleneck models.

Prior literature does not appear to have investigated the effect of label noise on the accuracy of concept prediction.

The finding that joint training maintains classification accuracy best while incurring a large drop in concept prediction accuracy is interesting.

**Weaknesses:**

It is unsurprising that injecting label noise reduces predictive performance, and it is known that sharpness-aware minimization improves performance when label noise is present.

The findings are purely empirical, and it is unclear how representative the noise models used in the paper are for the types of noise that occur in real-world settings.

The classifier considered in the submission is a linear one, and it is unclear how the results are affected by the fact that this classifier is relatively insensitive to noise.

There is a section on label smoothing in the appendix. Label smoothing is not referred to in the main text.

A couple of other small issues:

"need to be trained with noisy labels" - rephrase

Multiple duplicate references in the bibliography.

**Questions:**

N/A

---

> ### Author Response · Authors · 2024-11-22
> **Response to To3J (1/2)**
>
> We really appreciate the reviewer’s positive and constructive feedback. While we address the reviewer’s specific comments below, we would be keen to engage in any further discussion.
>
> ---
>
> **Novelty**
>
> > It is unsurprising that injecting label noise reduces predictive performance, and it is known that sharpness-aware minimization improves performance when label noise is present.
>
> We appreciate the reviewer’s comments and agree that label noise may intuitively be expected to degrade model performance. However, quite surprisingly, our findings reveal that Concept Bottleneck Models (CBMs) demonstrate robustness to target noise, which contradicts initial assumptions, but are highly vulnerable to concept noise. To the best of our knowledge, we are the first to explicitly identify this phenomenon and analyze it through extensive experiments.
>
> Specifically, we uncover two key reasons for this vulnerability to concept noise:
> * First, concept noise disrupts representation clustering by preventing the model from forming distinct and meaningful clusters in the representation space, which undermines its interpretability and performance.
> * Second, concept noise distorts concept-target relationships by altering the learned correlations between concepts and the target variable, resulting in unreliable predictions.
>
> Lastly, we explore whether noise mitigation methods could address these points and enhance the robustness of CBMs. Focusing primarily on Sharpness-Aware Minimization (SAM), we find that even modest improvements in concept prediction lead to significant enhancement in target prediction performance.
>
> ---
>
> **Represent real-world noise**
>
> > The findings are purely empirical, and it is unclear how representative the noise models used in the paper are for the types of noise that occur in real-world settings.
>
> Thank you for raising this point. To address concerns about practicality in our noise setting, we begin by describing the **symmetric noise** used in this work. This randomly flips labels within each class to any other class label with equal probability. It has been extensively studied in prior works as a representative proxy for real-world scenarios [1, 2, 3].
>
> Additionally, the **pairwise noise** (i.e., asymmetric noise) used in our study provides a realistic representation of label noise by incorporating dependencies between labels. For instance, it accounts for the scenarios where confusing a dog with a cat is more likely than confusing a dog with a flower. This setting captures an additional aspect of real-world noise scenarios [4].

---

> ### Author Response · Authors · 2024-11-22
> **Response to To3J (2/2)**
>
> **Effect of linear model**
>
> > The classifier considered in the submission is a linear one, and it is unclear how the results are affected by the fact that this classifier is relatively insensitive to noise.
>
> In our study, we use a linear model following the original research [5]. For the goal of CBMs, i.e., interpretability, linear models can offer a transparent decision-making process. In contrast, while non-linear models can capture more complex patterns, they are more sensitive to label noise, increasing the risk of learning incorrect patterns. This can lead to predictions that appear accurate but rely on unreliable or incorrect concepts, ultimately undermining the interpretability, a key advantage of CBMs.
>
> We also experimented with a target classifier using non-linear models to evaluate the generality of our findings. The results demonstrated that **the difference between linear and non-linear models does not significantly impact our discoveries**. The vulnerability of CBMs to concept noise became even more pronounced with non-linear models, further strengthening our claims. The results are provided below.
>
> | noise | $f$ | $nr = 0\%$ | $nr = 10\%$ | $nr = 20\%$ | $nr = 30\%$ | $nr = 40\%$ |
> |:--------------:|:---------------:|:-----------:|:-----------:|:-----------:|:-----------:|:-----------:|
> | Concept | Linear | 74.3 | 61.9 | 50.3 | 33.1 | 4.0 |
> | Concept | 2 Layer | 74.0 | 60.1 | 48.0 | 25.8 | 6.3 |
> | Concept | 3 Layer | 72.5 | 55.8 | 45.0 | 21.0 | 5.4 |
> | Target | Linear | 74.2 | 74.6 | 74.4 | 74.6 | 74.4 |
> | Target | 2 Layer | 74.2 | 74.0 | 74.2 | 75.0 | 74.0 |
> | Target | 3 Layer | 72.1 | 73.2 | 73.6 | 74.3 | 73.6 |
>
> We summarize our findings as follows:
> * Under target noise conditions, non-linear classifiers demonstrate similar performance to single-layer linear models.
> * Under concept noise, performance degradation became more evident with higher model complexity.
>
> This finding underscores our claim that "CBMs are highly vulnerable to concept noise," and this vulnerability becomes even more pronounced with non-linear models, further supporting our argument.
>
> ---
>
> **Marginal issues**
>
> > There is a section on label smoothing in the appendix. Label smoothing is not referred to in the main text.\
> > A couple of other small issues: "need to be trained with noisy labels" - rephrase\
> > Multiple duplicate references in the bibliography.
>
> Thank you for your suggestion. We will reflect it to our final version.
>
> ---
>
> [1] Wang, Yisen, et al. “Symmetric cross entropy for robust learning with noisy labels.” ICCV, 2019. \
> [2] Chen, Pengfei, et al. “Understanding and utilizing deep neural networks trained with noisy labels.” ICML, 2019. \
> [3] Lukasik, Michal, et al. “Does label smoothing mitigate label noise?” ICML, 2020. \
> [4] Scott, Clayton, et al. “Classification with asymmetric label noise: Consistency and maximal denoising.” JMLR, 2013. \
> [5] Koh, Pang Wei, et al. “Concept bottleneck models.” ICML, 2020.

---

> > ### Comment · Reviewer_To3J · 2024-11-25
> > **Response to authors' comments**
> >
> > Thank you for posting the additional results obtained using multi-layer networks. It seems important to also consider the size of the layers and how training was performed (i.e., whether some mechanism was employed to prevent overfitting).
> >
> > It is noteworthy that estimated target accuracy actually slightly increases with increased levels of target noise (up to nr = 30).

---

> > > ### Author Response · Authors · 2024-11-26
> > > **Thank you**
> > >
> > > Thank you for your thoughtful feedback. Your suggestion to conduct non-linearity experiments was invaluable in reinforcing our claims, uncovering new insights into CBMs, and enhancing the clarity and precision of our manuscript. We sincerely appreciate your guidance and effort.
> > >
> > > We are grateful that the reviewer finds our work sound and noteworthy for practitioners. Given the reviewer’s positive reviews and scores for soundness, presentation, and contribution (3,3,2), and provided that our response is adequate, we hope that the reviewer could reconsider escalating the initial rating of this work.

---

### Meta-Review · Area_Chair_SDZC · 2024-12-21

**Metareview:**

The paper investigates the impact of label noise on Concept Bottleneck Models (CBMs). The Authors perform intensive empirical studies. Based on the insights from these studies, they propose to use Sharpness-Aware Minimization (SAM) to improve CBMs.

The Reviewers appreciate the extent of empirical studies. Nevertheless, they underline the lack of theoretical insights, point out possible extensions of the experiments (e.g., empirical demonstration that label noise exists in output from LLMs and VLMs), and limited references to related work. The general observation that noise impacts performance is indeed not surprising. Finding a remedy of this problem is certainly very welcome. Unfortunately, the Reviewers are not fully convinced by the solution proposed by the Authors.

**Additional Comments On Reviewer Discussion:**

The Authors delivered additional experimental studies and tried to justify their finding theoretically. Nevertheless, the majority of the Reviewers remain unconvinced by the Authors.

---

### Decision · Program_Chairs · 2025-01-22

Reject